# Cold-induced nucleosome dynamics linked to silencing of Arabidopsis *FLC*

**Miguel Montez** [1,8], **Danling Zhu** [1,5,8], **Jan Huertas** [2,3], **M. Julia Maristany** [2,4], **Bas Rutjens** [1,6], **Mathias Nielsen** [1,7], **Rosana Collepardo-Guevara** [2,3,4] & **Caroline Dean** [1] ✉

Temperature influences nucleosome dynamics, and thus chromatin, to regulate gene expression. Such mechanisms underlie the epigenetic silencing of Arabidopsis *FLOWERING LOCUS C* (*FLC*) by prolonged cold. Here, we show a temperature-dependent transition in local chromatin structure at the H3K27me3 nucleation region, from a modality active for transcription to a state that can be Polycomb silenced. In vivo chromatin measurements and coarse-grained simulations at near-atomistic resolution show that the active transcription state is characterised by a highly dynamic nucleosome arrangement that exposes the *FLC* transcription start site (TSS). Cold exposure then changes the chromatin by reducing nucleosome dynamics and re-positioning the + 1 nucleosome, leading to transcriptional repression. This local chromatin transition partially depends on VERNALIZATION1 (VRN1), a non-sequence-specific DNA-binding protein. Loss of VRN1 results in hyperaccumulation of H2A.Z, more dynamic nucleosomes and an inability to accumulate H2Aub and H3K27me3. Our work highlights how local nucleosome dynamics link to chromatin structure transitions to integrate temperature inputs into epigenetic switching mechanisms in plants.

Histone modifications and variants[1–6], DNA supercoiling[7], RNA-DNA hybrids[8,9], and the intrinsic plasticity of nucleosome particles[10–17] all influence the biological function of chromatin. However, how local structural alterations in chromatin interact with these factors to impact gene regulation is still poorly understood. One gene where the impact of different regulatory mechanisms has been integrated is Arabidopsis *FLOWERING LOCUS C* (*FLC*). This is a target of many conserved co-transcriptional and epigenetic pathways and so has become an important model for understanding how local chromatin influences gene silencing. Epigenetic silencing of the *FLC* gene by the prolonged cold of winter promotes flowering; this aligns the developmental

transition to reproduction with the favourable environmental conditions of spring[18–20]. *FLC* is transcriptionally repressed during cold exposure and switched to an epigenetically silenced state by a Polycomb Repressive Complex 2 (PRC2). The transcriptional repression occurs gradually and involves antisense transcription-coupled demethylation of H3K36me3 and H3K4me1[21–23]. In contrast, PRC2 nucleation occurs in an ON/OFF switch-like mode independently at each allele, resulting in H3K27me3 at the Polycomb Response Element (PRE)/nucleation region (the first -three nucleosomes of the gene). At the whole organism level, this manifests as slow, quantitative, local accumulation of H3K27me3[24–27]. Once plants return to warm temperatures,

[1]John Innes Centre, Norwich Research Park, Norwich, United Kingdom. [2]Yusuf Hamied Department of Chemistry, University of Cambridge, Cambridge, United Kingdom. [3]Department of Genetics, University of Cambridge, Cambridge, United Kingdom. [4]Department of Physics, University of Cambridge, Cambridge, United Kingdom. [5]Present address: Key Laboratory of Molecular Design for Plant Cell Factory of Guangdong Higher Education Institutes, Institute of Plant and Food Research Department of Biology, School of Life Sciences, Southern University of Science and Technology, Shenzhen, China. [6]Present address: Spark Genetics, Padualaan 8, 3584 CH, Utrecht Science Park, Utrecht, The Netherlands. [7]Present address: Department of Immunology and Regenerative Biology, Weizmann Institute of Science, Rehovot, Israel. [8]These authors contributed equally: Miguel Montez, Danling Zhu. ✉e-mail: caroline.dean@jic.ac.uk

H3K27me3 spreads over the entire locus in a cell cycle-dependent fashion, delivering effective long-term silencing. Many other histone modifications have been characterised at *FLC*, as well as other cold-induced changes in the chromatin state but how they integrate to deliver epigenetic silencing remains unclear.

Although *FLC* silencing must be stable through cell division to provide a memory of winter, *FLC* must be reactivated after flowering to ensure the next generation of plants requires vernalisation. This reactivation occurs in the embryo and, potentially, also in the gametes[28,29]. This requires the SWR1 remodelling complex, which exchanges H2A with the variant H2A.Z[30–33]. Mutants in components of the SWR1 complex display early flowering phenotypes[30,31] and a compromised response to temperature[34]. H2A.Z has previously been shown to enable a dynamic response to temperature changes[34,35], but its impact on Polycomb silencing is not fully understood.

The components required for *FLC* silencing have been elucidated using forward genetic screens. These identified *VRN1*, *VRN2*, *VIN3* and *VRN5* and except for *VRN1*, these encode components of the core PRC2 and PRC2-accessory proteins[36]. VRN1 contains two DNA-binding domains with characteristics of the plant-specific B3 transcription factor family[37]. VRN1 binds DNA in a non-sequence-specific manner in vitro and broadly associates with all five metaphase chromosomes in vivo[37,38]. Surprisingly, we find that loss of VRN1 leads to local over-accumulation of H2A.Z at *FLC*. This led us to hypothesise that VRN1 affects nucleosome dynamics, contributing to the silencing of *FLC*. We show that VRN1 mediates cold-induced changes to the nucleosomes at the *FLC* nucleation region, including repositioning of the +1 nucleosome. We also show that VRN1 associates with chromatin just upstream of the *FLC* transcription start site (TSS), interacts in vivo with a range of chromatin remodelers and histone deacetylases, and promotes PRC1 and PRC2 activity at the *FLC* nucleation region. This encouraged us to investigate the structural behaviour of chromatin around *FLC* using molecular simulations with a near-atomistic residue/base pair-resolution coarse-grained model[13]. Combined, our data show how nucleosome dynamics link to local chromatin structure transitions to integrate temperature inputs into epigenetic switching mechanisms.

## Results

### Polycomb nucleation at *FLC* requires VRN1-mediated chromatin regulation

To investigate the role of VRN1 in *FLC* silencing, we analysed *FLC* mRNA levels over a cold exposure time course comparing wild-type FRI (a Col-0 background containing a functional *FRIGIDA* allele) and *vrn1-4* FRI. The *vrn1* mutation compromises *FLC* silencing, as previously observed in a different Arabidopsis genotype[37]. Moreover, the effect of *vrn1* on *FLC* expression was as strong as the PRC2-accessory protein mutant *vin3* (Fig. 1A). The *vrn1* mutation also prevented the quantitative accumulation of the PRC2-mediated H3K27me3 and PRC1-mediated H2Aub histone marks that are observed at the nucleation region in wild-type plants (Fig. 1B, C).

Chromatin IP analysis showed the histone variant H2A.Z hyper-accumulates within the nucleation region of *FLC* in the *vrn1* mutant, both before and after cold (Fig. 1D). This suggested that VRN1 function is associated with removing or limiting H2A.Z from *FLC* chromatin. ARP6 is a component of the SWR1 remodelling complex involved in the deposition of H2A.Z in Arabidopsis and an *arp6* mutation affects the levels of H3 acetylation, H3K4me3[39] and H2A.Z[33] over the promoter-proximal region of *FLC*, lowering *FLC* expression and accelerating flowering[31]. Genetic analysis between *vrn1-4* FRI and *arp6-1* FRI showed that the late flowering phenotype of *vrn1-4* FRI is suppressed by *arp6-1* after plants returned to warm conditions (Supplementary Fig. 1A, B), with *FLC* expression levels in the double mutant very similar to those in the single *arp6-1* FRI (Supplementary Fig. 1C). This supports a role for VRN1 in removing H2A.Z from *FLC* chromatin during epigenetic silencing.

### VRN1 is a chromatin regulator that associates with *FLC* 5′ region

Next, we complemented the *vrn1* mutant with a transgene containing a FLAG-tagged VRN1 genomic sequence under its endogenous 5′ and 3′ regulatory sequences. VRN1-FLAG *vrn1* FRI rescues the WT FRI flowering phenotype (Supp Fig. 1D). ChIP experiments on two independent VRN1-FLAG *vrn1* FRI transgenic lines showed high levels of VRN1 protein accumulating at the 5′ region of *FLC* (Fig. 2A). This indicates that even though VRN1 can bind DNA in a non-sequence-specific manner[37,38], it preferentially accumulates close to the *FLC* TSS in vivo. Surprisingly, VRN1 association with *FLC* chromatin is observed both before and after cold at similar levels (Fig. 2A). These results could explain the constitutive accumulation of H2A.Z at *FLC* in the *vrn1* mutant.

To gain insight into the function of VRN1 in *FLC* chromatin regulation, we performed IP-MS using two independent transgenic lines of the VRN1-FLAG *vrn1* FRI. Proteins associated with VRN1-FLAG were immunoprecipitated from purified nuclei from crosslinked seedlings grown at warm temperature, and identified by mass spectrometry (Fig. 2B). WT FRI plants without the VRN1-FLAG transgene were used as a control for the IP and only proteins enriched in the VRN1-FLAG IP relative to WT IP were considered. Gene ontology analysis showed that the enriched proteins are overrepresented in cellular component (CC) terms associated with chromatin remodelling and histone deacetylation (Fig. 2C). Among the most prevalent interactors of VRN1 are components of histone deacetylase complexes (HDACs) and chromatin remodelling complexes from the INO80 and SWI/SNF families (Fig. 2D). Both PRC1 and PRC2 components were also enriched in the VRN1 IP (Fig. 2D). Many of the interactions were also detected in IP-MS from plants subjected to three weeks of cold (Supp Fig. 2).

### VRN1-mediated cold-induced silencing of *FLC* involves alterations to the nucleosomes

Since VRN1 impacts *FLC* chromatin and interacts with chromatin regulators, we used Fluorescence Recovery After Photobleaching (FRAP) to measure the impact of VRN1 on histone dynamics. Globally, H2A.Z-containing chromatin, assayed using an HTA9-GFP transgene[34] in *vrn1*, was found to be more dynamic than in WT (Fig. 2E). H3.3-containing chromatin (assayed using HTR5-GFP[40] is also more dynamic in *vrn1* than WT (Supplementary Fig. 3A). At *FLC*, the H3.3 levels are changed in *vrn1* only in plants subjected to cold (Supplementary Fig. 3B). In contrast, H2A.Z changes in *vrn1* are observed both in the warm and cold, suggesting a different impact of *vrn1* mutation on the two histone variants. These results support the idea that VRN1 promotes a less dynamic chromatin state, consistent with an opposing function to ARP6 activity, as the genetic analysis had indicated. The FRAP analysis also supports a general function for VRN1 in chromatin dynamics, consistent with the widespread association of VRN1 with mitotic chromosomes, and the regulation of genes other than *FLC* independently of vernalisation[37,38]. Using a dominant repressor tag to overcome genetic redundancy, VRN1 was shown to be essential for Arabidopsis development[41].

We then explored the role of VRN1 in chromatin dynamics at *FLC*. We selected an approach to investigate nucleosome dynamics based on salt-solubility of micrococcal nuclease (MNase)-treated chromatin, shown to reflect epigenome dynamics[42]. After MNase digestion of purified and permeabilised Arabidopsis nuclei, low salt buffer was used to solubilise and extract chromatin. DNA was assayed using five different qPCR amplicons, all mapping in the 5′ region of *FLC*. The results show higher levels of solubilised chromatin before cold compared to after, and in the *vrn1* compared to WT (Fig. 3A). This suggests that biophysical properties of the nucleosomes are altered by cold and VRN1 in a similar manner, potentially with the accumulation of H2A.Z accounting for the increased chromatin solubility and higher nucleosome dynamics in the *vrn1* mutant.

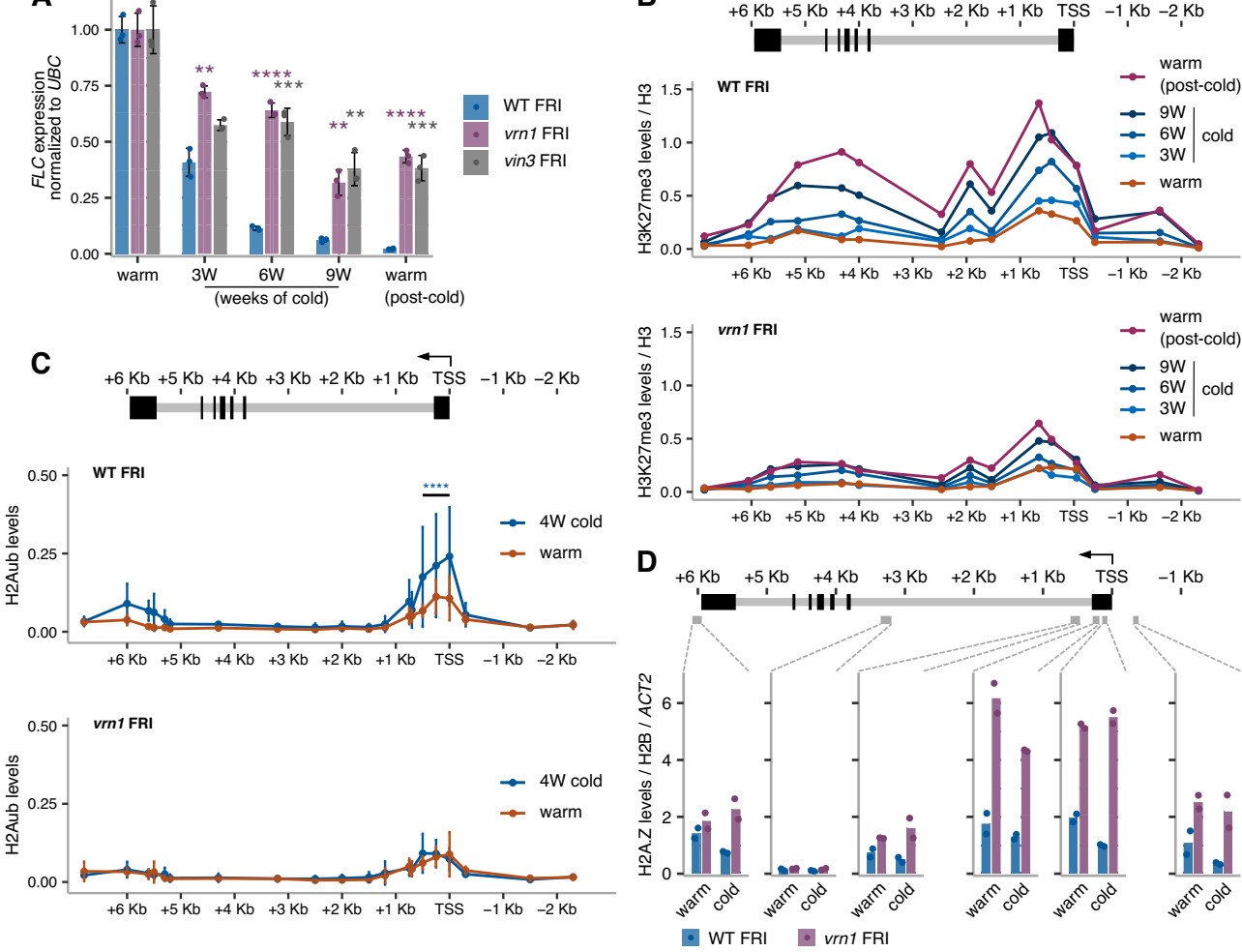

**Fig. 1 | VRN1 is required for the cold-induced transcriptional shutdown and Polycomb nucleation at *FLC*. A–C** Seedlings grown at warm temperature, or subjected to three (3 W), four (4 W), six (6 W) or nine (9 W) weeks of cold, and 10 days back to warm after nine weeks of cold (warm post-cold). (A) *FLC* mRNA levels by RT-qPCR normalised to *UBC21* (AT5G25760) in WT FRI, *vrn1-4* FRI, and *vin3-4* FRI, in the cold and post-cold relative to the warm. **B** H3K27me3 ChIP-qPCR levels as percent of input normalised to H3 in WT FRI and *vrn1-4* FRI. **C** H2Aub ChIP-qPCR levels in percent of input normalised to H3 in WT FRI and *vrn1-4* FRI seedlings grown in the warm and subject to 4 weeks of cold. Statistical differences were assessed over a region encompassing three amplicons. **D** H2A.Z ChIP-qPCR levels in percent of input normalised to H2B and *ACTIN2* in WT FRI and *vrn1-4* FRI seedlings in the warm and after six weeks of cold. **A–D** Bars and error bars show the mean +/− SD. Points show individual biological replicates, *n* = 3 (**A**, **C**), *n* = 2 (**B**, **D**). **\*\***p-value < 0.01, \*\*\*p-value < 0.001, \*\*\*\*p-value < 0.0001 from two-tailed Student's *t*-test. Schematic representation of the *FLC* locus on the top of the plots with exons as black boxes and introns as grey lines. An arrow at the TSS indicates the direction of transcription.

## High-resolution mapping of the nucleosome distribution at *FLC*

To understand which aspect of the biophysical properties of the nucleosomes was changing in response to cold and VRN1 we performed a more comprehensive analysis of nucleosome dynamics. We used a targeted MNase-seq to map nucleosomes on genes of interest. Chromatin was digested with MNase in Arabidopsis nuclei, and subsequently, mono- and sub-nucleosomal DNA fragments were isolated and used for whole-genome library preparation. These libraries were then specifically enriched by bait hybridisation for the DNA sequences from *FLC* and other selected loci and then paired-end sequenced (see "Methods"; Fig. 3B). This targeted enrichment allowed high-resolution mapping of nucleosomes at *FLC*. The full nucleosomal DNA sequences (length 100-150 bp) were reconstituted from paired reads and plotted along the *x*-axis displaying positional information (Fig. 3C). DNA fragment length changes, plotted on the *y*-axis, are indicative of biophysical differences between nucleosomes and/or their digestion[43–45]. The nucleosome distribution at *FLC* (Fig. 3C) has a typical short nucleosome-depleted region (NDR) around the *FLC* TSS, flanked by prominent nucleosomes at −1 and +1, as generally observed in most

eukaryotes including Arabidopsis[46–48]. The nucleosomes across the entire *FLC* gene show a diverse pattern of positions and DNA length indicative of a variety of nucleosome states. Well-defined positions are detected at the borders of the gene (both 5' and 3' end), partially agreeing with the predicted nucleosome positioning based on the genomic DNA sequence (Fig. 3C, and Supplementary Fig. 4A). Surprisingly, despite the changes in *FLC* expression in response to cold, there were no large-scale changes in nucleosome occupancy at *FLC* after normalisation to the total occupancy at reference genes (Supplementary Fig. 4B).

## Cold temperature drives the repositioning of the +1 nucleosome closer to *FLC* TSS

A closer inspection of the nucleosomes near *FLC* TSS in warm-grown plants (Fig. 3D) revealed the presence of either a di-nucleosome structure, with the +1 and +2 nucleosomes frequently occupying adjacent positions, or one nucleosome that could occupy alternative positions. The MNase-protected fragments of 250-300 bp in that region favour the di-nucleosome model (Supp Fig. 4C). Moreover, the

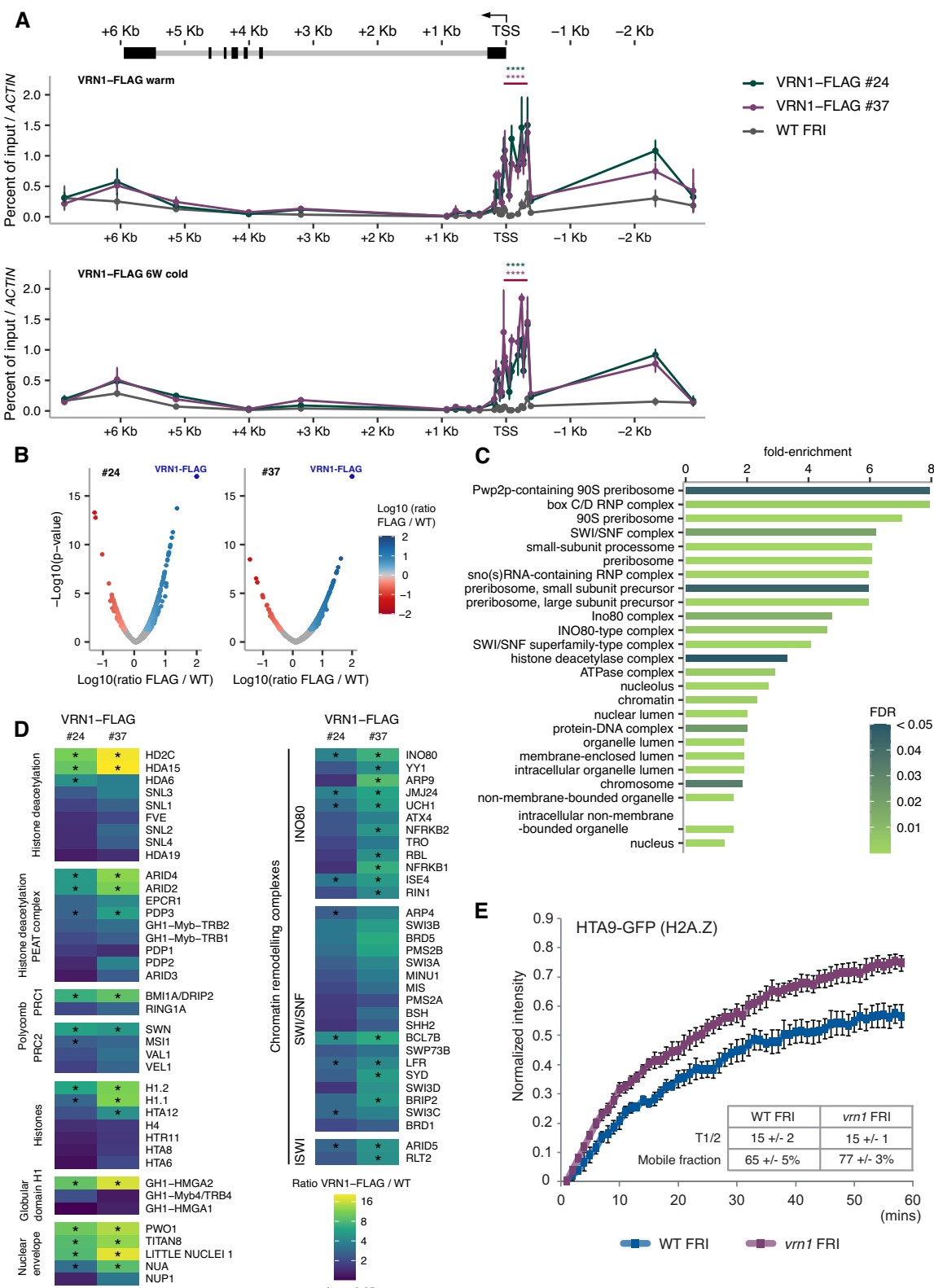

MNase-seq data in plants exposed to cold shows repositioning of the +1 nucleosome closer to *FLC* TSS; and this is attenuated and more variable in the *vrn1* mutant (Fig. 3D, E, and Supplementary Fig. 5, 6). Previous findings had reported a stabilisation of the *FLC* +1 nucleosome upon cold exposure[49]. However, we now interpret this as different PCR amplicon efficiency in the detection of the +1 nucleosome before and after repositioning; only half of the forward primer used

overlaps with the +1 nucleosomal DNA in the warm while it fully overlaps the +1 DNA in the cold. The cold-induced repositioning of the +1 nucleosome is specific to *FLC* as we did not observe the same effect on other genes tested (Supplementary Fig. 7A–D). Nucleosome positioning has a known impact on gene expression in eukaryotes[50,51]. We suggest a model (Fig. 4) where cold-induced *FLC* silencing involves changes to nucleosome dynamics that include repositioning of the +1

**Fig. 2 | VRN1 controls *FLC* chromatin dynamics. A** FLAG ChIP-qPCR levels as percent of input in the VRN1-FLAG *vrn1-4* FRI #24 and #37 normalised to *ACT7* (AT5G09810) and relative to WT FRI. Seedlings grown at warm temperatures (top panel) or subjected to 6 weeks of cold (6 W cold; bottom panel). Points and error bars show the mean +/− SD, $n = 3$ biological replicates. ****$p$-value < 0.0001 from two-tailed Student's $t$-test over a region encompassing eight amplicons. Schematic representation of the *FLC* locus on the top of the plot with exons as black boxes and introns as grey lines. An arrow at the TSS indicates the direction of transcription. **B** Volcano plot with all proteins identified by mass spectrometry analysis after VRN1-FLAG IP in the VRN1-FLAG *vrn1-4* FRI #24 (left) and #37 (right) transgenic seedlings grown at warm temperature. Colour scale indicates the enrichment ratio of FLAG IP in transgenic plants over IP in plants without the VRN1-FLAG transgene (on a logarithmic scale), $p$-value from two-tailed Student's $t$-test. **C** Gene ontology terms for cellular component (GO:CC) for the proteins from VRN1-FLAG IP-MS for which the enrichment ratio over IP in plants without the VRN1-FLAG transgene was higher than 2-fold and $p$-value lower or equal to 0.05, in both lines #24 and #37, $p$-value from Fisher's Exact test with Bonferroni correction for multiple testing. **D** Proteins enriched in the VRN1-FLAG IP over IP in plants without the VRN1-FLAG transgene involved in chromatin regulation selected from the GO:CC terms. The colour scale indicates the enrichment levels on a logarithmic scale. *$p$-value < 0.05 from two-tailed Student's $t$-test. **E** FRAP normalised fluorescence intensity for pHTA9::HTA9-GFP (H2A.Z) in WT FRI and *vrn1-4* FRI seedlings grown at warm temperature. Points and error bars show the mean +/− SD, $n = 5$ biological replicates. The average half-time recovery (T1/2) and percentage of mobile fraction for each genotype are shown.

nucleosome towards the TSS, and this precedes Polycomb activity at the locus. VRN1 plays a role in this mechanism by limiting H2A.Z incorporation, which is necessary for both the *FLC* repression and the activities of PRC1 and PRC2 at the nucleation region.

## Modelling the chromatin dynamics at the *FLC* nucleation region

Our model suggests that there are cold-induced VRN1-mediated changes in nucleosome dynamics and chromatin structure locally at the *FLC* nucleation region, linked to silencing. To further understand this, we employed our chemical-specific coarse-grained modelling approach that has been used to investigate the link between different aspects of the nucleosomes and the 3D structural dynamics of chromatin[13]. Specifically, a series of Debye-length Hamiltonian replica exchange molecular dynamics (MD) simulations were performed (see "Methods") on four-nucleosome chromatin arrays mimicking the *FLC* 5′ end. These arrays include the −1 nucleosome, the NDR (where the TSS is located), and the first three nucleosomes of the gene. Unless stated otherwise, nucleosome breathing and sliding motions were allowed; herein referred to as "plastic" chromatin. In the coarse-grained model, these nucleosome motions are dictated by the relative strength of DNA–histone core electrostatic interactions (which promote nucleosome stability) and the sequence-dependent mechanical rigidity of the nucleosomal DNA (which inhibits nucleosome stability). The simulations characterise the equilibrium configurational ensembles of *FLC* chromatin and thus, how 3D chromatin structure is influenced by nucleosome dynamics.

We use the nucleosome positions mapped with MNase-seq in vivo to build two different chromatin models. One model with nucleosomes initially positioned as most frequently observed in plants in the warm–"warm positioning" (Fig. 5A, and Supplementary Fig. 8A); and another model with nucleosomes initially positioned as most frequently observed in plants subjected to cold–"cold positioning" (Fig. 5B, and Supplementary Fig. 8B). The cold positioning is more robustly maintained during the simulations (Supplementary Fig. 8A, B), indicating a mechanical structural preference for those positions. We then assessed overall chromatin compaction based on the radius of gyration of the simulated chromatin equilibrium ensembles, which measures the spatial distribution of nucleosomes around the chromatin centre of mass. The results show that the radius of gyration is higher in chromatin with warm positioning (Fig. 5C). This suggests that the nucleosome positions observed experimentally in the warm give rise to more open structures of *FLC* chromatin over the entire 5′ end region. In such structures, the DNA around the TSS is more readily exposed, thus consistent with higher transcription.

We further explored *FLC* chromatin structure through a pairwise analysis of the physical distances between all nucleosomes. The results show that in chromatin with warm positioning, the −1 nucleosome remains more distant from the rest, compared to cold positioning (Fig. 5D). Remarkably, this is linked to a more compact 3D structure locally around the region encompassing the nucleosomes +1, +2, and +3 (Fig. 5D). This structural behaviour exposes the *FLC* TSS and

maintains an overall less compact 3D chromatin structure (Fig. 5A and C). The cold-induced nucleosome repositioning disfavours the compact local structure, bringing the −1 nucleosome closer to the rest (Fig. 5B, D). This reduces the accessibility of the TSS whilst increasing the accessibility of the first three nucleosomes covering the PRC2 nucleation region. We speculate this may promote the switch of *FLC* chromatin structure from transcriptionally active to a PRC2 silenced state.

Next, we explored the contribution of chromatin plasticity to these states. This was done by preventing nucleosome sliding and DNA unwrapping (introducing stiff harmonic bonds between the 147 base pairs of the DNA and the histone complex), referred to as "rigid" chromatin in contrast to "plastic". Although constraining nucleosome plasticity did not change the median chromatin compaction, a broader ensemble of conformations is observed in simulations of rigid chromatin with warm positioning (Supplementary Fig. 8C), whereas the opposite is observed in chromatin with cold positioning (Supplementary Fig. 8D). This suggests that nucleosome plasticity may be important to increase compaction variability in the cold. This is consistent with the MNase-seq data showing a different but still variable nucleosome positioning in vivo in the cold, relative to the warm (Supp Fig. 5, 6).

When using the "warm positioning" or "cold positioning", the simulations agree with the experimental data and help understand *FLC* chromatin structure dynamics. However, alternative interpretations are possible. For example, the presumed +1 and +2 positions could be two alternative positions for one nucleosome. Therefore, we simulated *FLC* chromatin with only three nucleosomes spaced by 260 and 52 bp, instead of four (Supplementary Fig. 9A–D). The ensembles exhibited much lower compaction levels (Supplementary Fig. 9D) and had an extended NDR (Supp Fig 9A-C) that is not consistent with the MNase-seq data in vivo. This supports the proposed "warm" and "cold positioning", and the cold-induced changes in 3D chromatin structure at *FLC*.

Given the role of VRN1 in the nucleosome dynamics at *FLC*, we sought to simulate a chromatin structure corresponding to a *vrn1* mutant, which accumulates the H2A.Z variant at the 5′ end (Fig. 1D). Therefore, we replaced the canonical H2A histones with the H2A.Z variant in the +1 and +2 nucleosomes in the simulations with cold-positioned nucleosomes (Supplementary Fig. 10A–D). The incorporation of H2A.Z increases the nucleosome dynamics at the *FLC* (Supp Fig 10C), which in turn increases the overall chromatin compaction (Supplementary Fig. 10D). Even though this might not be intuitive, MD simulations have elucidated how chromatin with high nucleosome dynamics manifests a liquid-like behaviour yet a higher degree of compaction[13]. Notably, two distinct configurations were observed in our simulations, both with the +1 nucleosome adjacent to the +2 and a compact +1/+2/+3 local structure, resembling chromatin with warm positioning (Supplementary Fig. 10A, B, D). This is consistent with H2A.Z disrupting the nucleosome positions in the cold. These results support that the high nucleosome dynamics observed in the *vrn1*

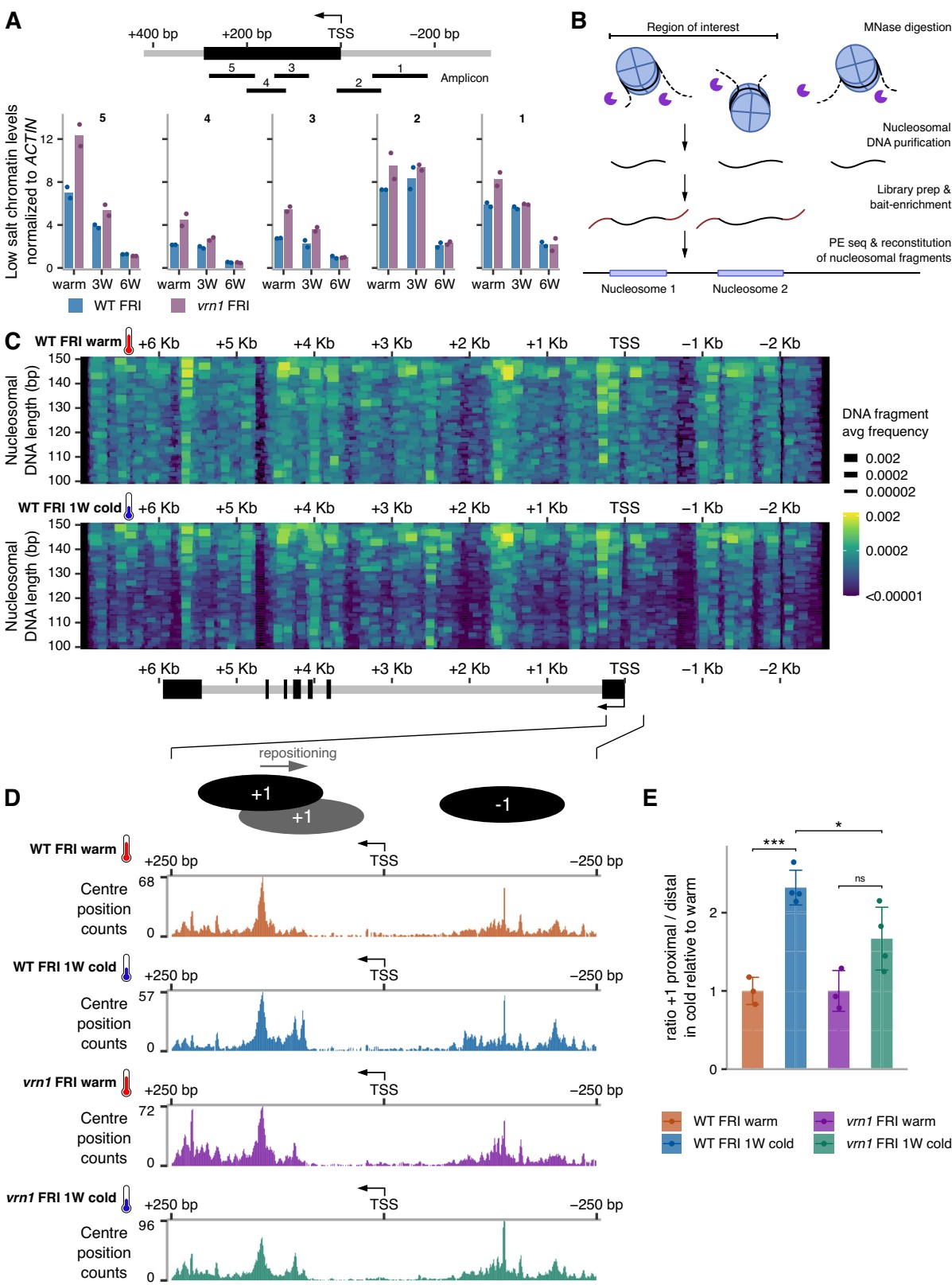

mutant are at least partially promoted by H2A.Z incorporation, and that limiting H2A.Z is important for the cold-induced *FLC* 3D chromatin structure dynamics.

## Discussion

In this study, we explored the link between nucleosome dynamics and transcriptional repression/epigenetic silencing of *FLC* in response to

temperature, and the role of VRN1 in that mechanism. Altogether, our results suggest that VRN1 contributes to *FLC* silencing through changes to the local nucleosome dynamics, which are required for the switch to the Polycomb-silenced state. The transient accumulation of H2Aub and the slower and quantitative accumulation of H3K27me3 at the *FLC* nucleation region are both compromised in the *vrn1* mutant, which hyperaccumulates the H2A.Z variant within the nucleation

**Fig. 3 | Nucleosome dynamics underlying the cold-induced silencing of *FLC*.**
**A** MNase-qPCR analysis at *FLC* on chromatin fraction eluted under low salt (80 mM NaCl) in WT FRI and *vrn1-4* FRI seedlings grown at warm temperature or subjected to 3 (3 W) or 6 (6 W) weeks of cold. qPCR was performed with 5 different primer pairs along the 5′ region of *FLC* (amplicons shown in the schematics above). Bars and error bars show the mean +/− SD. Points show independent biological replicates, *n* = 2. Data was normalised to undigested DNA and *ACTIN2*. Schematic representation of the 5′ end of *FLC* locus on the top of the plot with exon 1 in black and an arrow over the TSS indicating the direction of transcription. **B** Schematic representation of MNase-seq procedure where MNase was used to digest chromatin in purified nuclei. The DNA fragments protected against digestion were purified and used for library prep. Fragments from genomic regions of interest that include FLC and other loci were enriched from the genome-wide libraries to increase depth. MNase-protected DNA fragments are reconstituted from paired-end reads allowing high-resolution mapping of nucleosome positions. **C** The entire nucleosomal DNA fragments were plotted as coloured horizontal segments. The genomic position on the *x*-axis is aligned and in scale with the schematic representation of the *FLC* gene on the bottom; exons in black, introns in grey. The *y*-axis shows the length of the DNA fragments protected from MNase digestion and reconstituted from paired reads. Both segment colour and width scales represent the average frequency of a detected DNA fragment within the *FLC* locus in WT FRI seedlings at warm temperature (top), or after 1 week of cold (bottom). **D** Plot showing the centre position of nucleosomal DNA fragments reconstituted from paired MNase-seq reads at the 5′ end of *FLC* in WT FRI and *vrn1-4* FRI seedlings at warm temperature or after 1 week of cold (1 W cold). A representation of the +1 and −1 nucleosome positions inferred from the data were added to the schematics on the top. **E** *FLC* +1 nucleosome repositioning assessed by plotting the ratio of nucleosomal fragments within a TSS-proximal window divided by those in a TSS-distal window for each sample in the cold normalised to warm (see "Methods"). The windows are shown at the bottom of Supp5C. Bars and error bars represent the mean +/− SD. Points represent individual biological replicates, *n* = 3 for WT FRI warm and *vrn1-4* FRI warm, and *n* = 4 for WT FRI cold and *vrn1-4* FRI cold. ns *p*-value >= 0.05, **p*-value < 0.05, ***p*-value < 0.001 from two-tailed Student's *t*-test.

region (Fig. 1B-D). We also show that VRN1 functionality is no longer important for *FLC* silencing in an *arp6* mutant with already low H2A.Z, supporting H2AZ reduction as a consequence of VRN1 function. Our results are consistent with previous work suggesting that transient H2A.Z eviction mediated by heat shock factors is required for temperature-responsive genes to respond to high ambient temperature[35]. Moreover, previous genetics with *FVE*, a component of histone deacetylation complexes, has shown that ARP6 induces *FLC* expression via histone acetylation[39]. We found that histone deacetylation components are among the most predominant proteins associated with VRN1 from our IP-MS, including FVE (Fig. 2D). Components of chromatin remodelling complexes such as INO80 and SWI/SNF are also enriched in the VRN1 IP-MS (Fig. 2D). INO80 has been suggested to facilitate the exchange of H2A.Z variant by H2A[52–54]. Histone deacetylation and H2A.Z replacement may be involved in the VRN1-mediated chromatin state switch that opposes ARP6 function to silence *FLC*, yet this remains to be validated.

VRN1 strongly impacts the chromatin state at *FLC*, affecting both histone modifications and variants. We further explored other dynamic aspects of the chromatin at *FLC* in vivo using MNase-based methods. MNase has been extensively used not only to map nucleosomes but also to assess their biophysical properties[42–45,55–57]. Initially, we digested chromatin with MNase and extracted the chromatin fragments under low salt conditions and found both VRN1 and cold exposure reduce the chromatin solubility (Fig. 3A). We followed that up with a targeted sequencing approach from the entire fraction of MNase-digested chromatin and found that cold induces the repositioning of the *FLC* +1 nucleosome (Fig. 3D, E). In the *vrn1* mutant, the cold-induced repositioning of the +1 nucleosome is compromised and more variable. This supports the VRN1 role in limiting H2A.Z incorporation to reduce nucleosome dynamics and allow stable nucleosome repositioning in the cold. Simulations of *FLC* chromatin with H2A.Z agree with this model (Supplementary Fig. 10). The positioning of the +1 nucleosome has an important impact on the Pol II preinitiation complex assembly[50,58–60]. As the +1 nucleosome is closer to the TSS, transcription initiation is reduced, as recently characterised at a structural level[51]. The cold-induced repositioning of the +1 nucleosome towards the *FLC* TSS could therefore reduce *FLC* transcriptional initiation, contributing to the transcriptional shutdown necessary for PRC2 nucleation. What might lead to the repositioning of the +1 nucleosome? In yeast, non-coding antisense transcription invades the sense NDR, increasing H3K36me3 on the flanking nucleosomes, and these act as a recruitment platform for histone deacetylases. This is associated with nucleosome repositioning and narrowing of the NDR, which in turn reduces sense transcription[60,61]. We have previously shown that antisense transcription induced by cold extends into the *FLC* NDR[62], and this could be involved in the +1 repositioning and *FLC*

transcriptional shutdown. It will be important to explore this contribution of antisense transcription and its link to a possible VRN1-targeted histone deacetylation to the nucleosome repositioning and *FLC* shutdown.

A possible parallel mechanism arises from our in-silico MD simulations of chromatin around the *FLC* promoter and nucleation region. Although nucleosome positioning in the cold favours an overall more compact chromatin structure, the chromatin over the +1, +2, +3 nucleosomes that accumulate the H3K27me3 becomes more accessible (Fig. 5A–D). Thus, the +1-nucleosome repositioning at *FLC* might increase the local accessibility to PRC2 and/or enhance PRC2 deposition of H3K27me3 over these 3 nucleosomes. The dynamic ubiquitination of H2A is important for PRC2 silencing, but its function is not completely understood. H2Aub has been shown to prevent nucleosome stacking and, thus chromatin compaction, in vitro[63]. In the *vrn1* mutant, H2Aub fails to accumulate, and this is associated with defects in H3K27me3 nucleation, consistent with a role for local chromatin accessibility at *FLC* in promoting PRC2 activity. Moreover, the H2A.Z accumulation in the *vrn1* mutant may be acting synergistically with the lack of H2Aub favouring a compact local structure. The +1-nucleosome repositioning at *FLC* seems to play an important role in reducing this local compaction. This mechanism is also supported by work in *N. crassa* showing the repositioning of nucleosomes closer to the TSS by the ACF remodelling complex as a requirement for H3K27 methylation[64].

Our previous work has shown that the antisense-mediated transcriptional repression and the Polycomb silencing work in parallel rather than through a linear sequence of events controlling *FLC* expression outcome in response to cold[21]. The transcriptional repression pathway is able to repress *FLC* over both fast and slow timescales and increases the probability of each independent allele being silenced by the mechanistically independent Polycomb pathway. The local structural changes to *FLC* chromatin findings shown in this paper have the potential to influence both the transcriptional repression and Polycomb pathways. The repositioning of the +1 nucleosome driving changes to the local chromatin structure is observed within one week of cold, thus within the faster-acting transcriptional pathway. Characterisation over longer timescales will test the potential link to the slower Polycomb pathway. The observed high levels of H2A.Z, both before and after long periods of cold in the *vrn1* mutant, suggest that any chromatin structure alterations will persist through prolonged cold exposure. We propose that defective silencing of *FLC* in the *vrn1* mutant is associated with the accumulation of the H2A.Z variant, which enhances nucleosome dynamics and compromises the nucleosome positioning in the cold. The more variable nucleosome positions in simulations of H2A.Z-containing chromatin support the idea that H2A.Z incorporation increases chromatin plasticity, allowing for

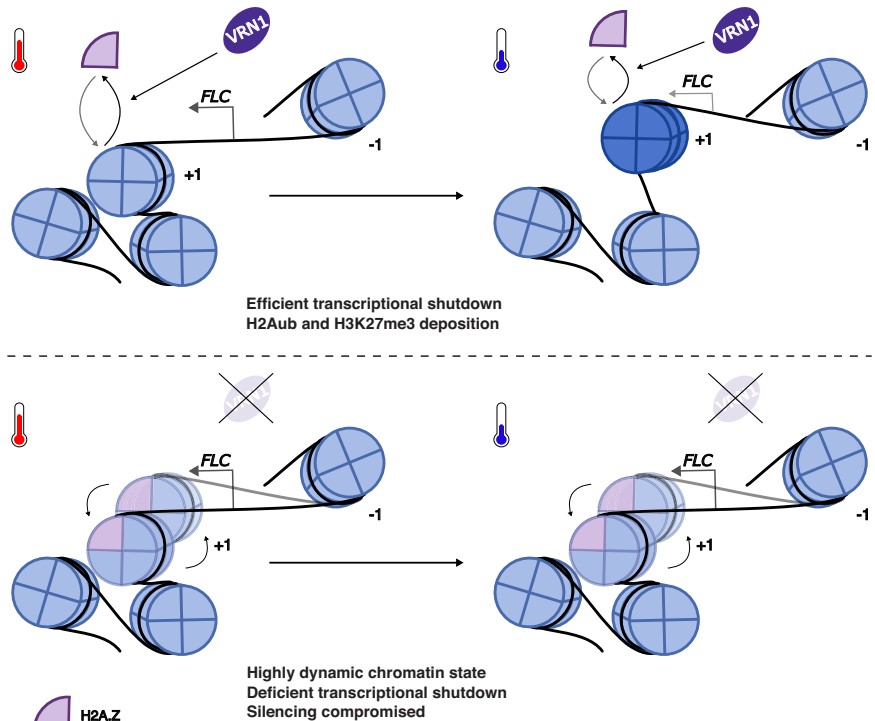

**Fig. 4 | Nucleosome dynamics modulating chromatin structure changes at *FLC* during cold-induced transcriptional silencing.** Schematic representation of the nucleosome dynamics at the 5' end of *FLC*. In the wild type (top), cold induces the repositioning of the +1 nucleosome towards the TSS with the potential to drive structural changes to the chromatin locally. This is associated with reduced nucleosome dynamics enabling transcriptional shutdown and epigenetic silencing of *FLC*. The local alterations to the chromatin are dependent on VRN1 possibly via limiting the incorporation of the histone H2A.Z variant. In a *vrn1* mutant (bottom), H2A.Z accumulates at the first nucleosomes of *FLC* increasing nucleosome dynamics and compromising the stable positioning of the +1 nucleosome.

transitions between different conformational states that interfere with stable Polycomb silencing. Generally, the simulations matched the experimental data well. However, we know that VRN1 binding/PRC2 activity may change the nucleosome structure and dynamics, and these effects were not considered in our MD simulations. Importantly, the fact that our simulations are based on the inherent nucleosome dynamics without additional protein factors suggests that nucleosome repositioning and H2A.Z alone change the local chromatin structure at *FLC*. Further analyses are needed to fully understand what drives the nucleosome repositioning important for *FLC* transcriptional repression and epigenetic switching. VRN1 is associated with *FLC* in the warm and cold, but VIN3 is a cold-induced protein that associates with *FLC* nucleation region and drives the switch to *FLC* epigenetic silencing[65,66]. The multivalent interactions between VIN3 and other PRC2-associated proteins and the VRN1-repositioning of the nucleosomes open an interesting future direction to explore the chromatin structure transitions at *FLC*.

Our data also raise the interesting possibility of specific 3D chromatin structures in vivo at *FLC*. Before cold, we most frequently detected MNase-resistant fragments of 140 bp and 110 bp for the +1 and +2 nucleosomes respectively (Fig. 3D). This particular DNA length footprint is consistent with a hexasome at the +2-nucleosome position. Moreover, the +1 and +2 nucleosomes appear to be adjacent or very near each other. This resembles a di-nucleosome structure referred to as an overlapping di-nucleosome (OLDN), where one partially unwrapped octasome is adjacent to a hexasome[67,68], frequently found at promoter-proximal regions[69,70]. The *FLC* promoter-proximal Polycomb nucleation region could be predominantly occupied by an OLDN structure in plants grown in warm temperatures. Upon transfer to the cold, the repositioning of the +1 nucleosome could resolve this structure. Our simulations suggest that the nucleosome positioning during active transcription in the warm generates a more compact local structure formed by the +1, +2 and +3 nucleosomes. While not intuitive, we speculate that this compact structure could hamper the deposition of H3K27me3 by the PRC2. However, it would not hamper RNA Pol II activity since the transcriptional machinery is equipped with various accessory proteins and chromatin remodelling complexes such as FACT, which aids transcription through nucleosomes maintaining chromatin integrity[71–73]. Other studies have characterised chromatin structures, including overlapping di- and tri-nucleosomes[69,74], telomeric columnar chromatin structure[75], centromeric chromatin[17,76,77], and other inter-nucleosome interactions[10,78–83]. We envisage that exploring these local chromatin structures and their inherent biophysical properties at a fine scale in the genome will significantly boost our understanding of gene regulation and epigenetic mechanisms in complex biological systems. Our work illustrates that the cold-induced silencing of *FLC* provides an excellent system for such studies.

## Methods

### Plant material

All genotypes used in this study have Columbia (Col-0) background with a functional *FRIGIDA* allele introgressed (recurrently back-crossed) from the ecotype San Feliu-2[84] referred to as WT FRI, generated by crossing *vin3-4* (SALK_004766.51.00.x[85]), *arp6-1* (Garlic_599_G0[31]), pHTA9::HTA9-GFP[34], and pHTR5::HTR5::GFP[40] with WT FRI. The *vrn1-4* FRI mutant obtained from fast-neutron mutagenesis[65] was back-crossed to WT FRI to confirm allelism. The VRN1-FLAG lines were generated by transforming *vrn1-4* FRI plants.

Seeds were surface-sterilised with chlorine gas, sown on plates with full-strength Murashige & Skoog (MS) medium including vitamins (Duchefa, M0222) plus 1% agar, and stratified for 3 days at 4 °C.

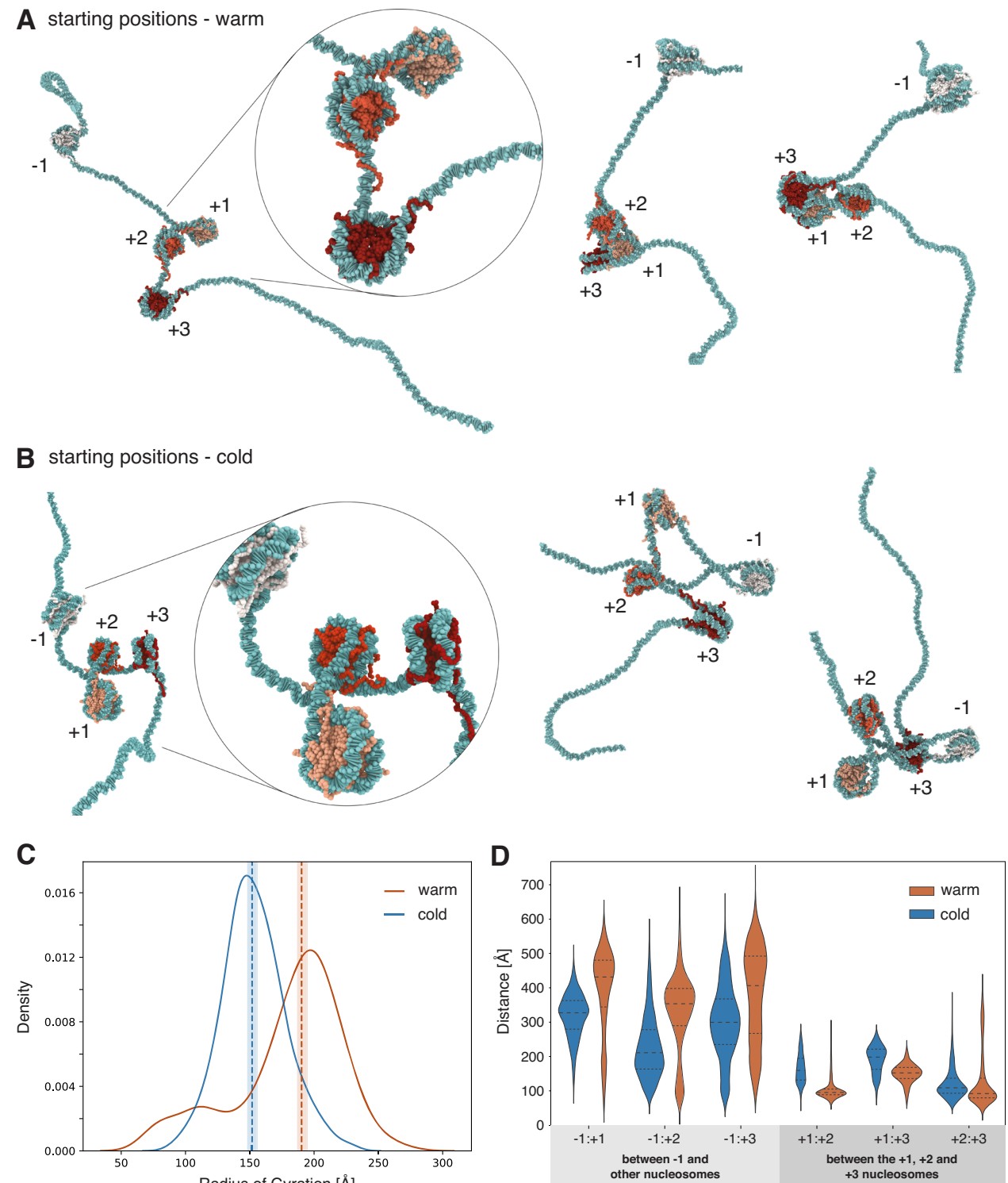

**Fig. 5 | Simulations reveal a nucleosome positioning-dependent chromatin structure. A** Representative snapshots of simulations of *FLC* chromatin at physiological salt concentration, either for nucleosomes with warm positioning or **B** cold positioning. **C** Distribution of the radii of gyration. **D** Distribution of inter-nucleosomal distances for the simulations of nucleosomes with warm positioning (orange) vs cold positioning (blue).

Seedlings were grown at 22 °C, 16 h light/8 h dark for 10 days – referred to in this paper as warm temperature/before cold. Seedlings were then subject to a cold treatment of several weeks (the number of weeks depends on the experiment and is indicated in Figure legends) before returning to warm temperature, referred to as warm post-cold. The vernalized plants were then transferred to soil to propagate seeds, crossing and phenotyping. Flowering time was assessed by counting

the total rosette leaf number. Plant material from seedlings before, during and post-cold was collected and stored at −80 °C for the experiments described below.

## Cloning
VRN1-FLAG was generated by cloning a genomic *VRN1* (AT3G18990) sequence from Col-0 including endogenous promoter, 5'UTR, 3'UTR,

and terminator (from 1,422 bp upstream the START codon to 506 bp downstream the STOP codon). The STOP codon was mutated to create a BamHI restriction site used for in-frame fusion of a Strep-II/FLAG TAP sequence at the C-terminus. The VRN1-FLAG fusion was then cloned into the binary vector pCambia1300 for plant transformation with *Agrobacterium tumefaciens* strain GV3101.

### Gene expression analysis by RT-qPCR

Total RNA was extracted using phenol-chloroform. A total 3 ug of RNA was treated with TURBO DNase I (Invitrogen) and used for cDNA synthesis with Superscript III (Invitrogen) at 52 °C for 1 h. The cDNA was diluted 25 times and used for qPCR (Roche, LightCycler 480) with SYBR Green Master Mix (Roche), and primers listed in Supplementary Data 1. The Ct values were normalised against the reference gene *UBC21* (AT5G25760).

### Chromatin immunoprecipitation (ChIP-qPCR)

ChIP experiments were performed as previously described[86]. Antibodies used in this study: Anti-GFP (Abcam, ab290), anti-FLAG M2 magnetic beads (Sigma, M8823), anti-H3 (Abcam, ab1791), anti-H3K27me3 (Millipore, 07-449), H2AK119ub (Cell Signalling Technology, #8240), anti-H2B (Abcam, ab1790). qPCR primers listed in Supplementary Data 1.

### Immunoprecipitation followed by mass spectrometry (IP-MS)

IP-MS was performed as previously described[22]. Briefly, seedlings from VRN1-FLAG *vrn1-4* FRI transgenic lines #24 and #37 were grown together with WT FRI (control) at warm temperature or three weeks after cold treatment. The plant tissue was crosslinked with 1% formaldehyde in 1X PBS for 10 min with vacuum. Two grams of crosslinked tissue were used per biological replicate. Two biological replicates ($n = 2$) for each genotype were used for IP-MS in the warm and three biological replicates ($n = 3$) were used for IP-MS in the cold. Nuclei were isolated with Honda buffer and purified on a Percoll gradient. The purified nuclei were treated with Benzonase and then sonicated $3 \times 5$ min (30 sec on/ 30 sec off) on Bioruptor standard (Diagenode) with 0.5% SDS. IP was performed overnight at 4 °C with 1.5 µg of anti-FLAG antibody (Sigma, F1804) coupled to 1.5 mg of M-270 epoxy Dynabeads (Invitrogen, 14311D) per IP reaction. Proteins were precipitated with chloroform:methanol (1.1:4.4). Mass spectrometry was performed as previously described[22] by the Mass Spectrometry service at the Proteomics facility at John Innes Centre with the following modifications: The samples were loaded onto a trap cartridge (PepMap™ Neo Trap Cartridge, C18, 5um, 0.3x5mm, Thermo) with 3% ACN in 0.1% TFA at 15 µl min-1 for 3 min. The trap column was then switched in-line with the analytical column for separation at 55 °C using the following gradient of solvents A (water, 0.1% formic acid) and B (80% acetonitrile, 0.1% formic acid) at a flow rate of 0.2 µl min-1: 0-3 min 3% B (parallel to trapping); 3-10 min linear increase B to 10%; 10-100 min linear increase B to 32%; 100-148 min linear increase B to 50%; followed by a ramp to 99% B and re-equilibration to 3% B, for a total of 180 min runtime. Mass spectrometry data were acquired with the FAIMS device set to three compensation voltages (−35V, −50V, −65V) at standard resolution for 1.0 s each with the following MS settings in positive ion mode: OT resolution 120 K, profile mode, mass range m/z 300–1800, normalised AGC target 100%, max inject time 50 ms; MS2 in IT Turbo mode: quadrupole isolation window 1 Da, charge states 2-5, threshold 1e4, HCD CE = 30, AGC target standard, max. injection time dynamic, dynamic exclusion 1 count for 15 s with mass tolerance of ±10 ppm, one charge state per precursor only. Raw data were processed and quantified in Proteome Discoverer 3.0 (Thermo). The CHIMERYS database search was performed with the inferys_2.1.0_fragmentation prediction model, a fragment tolerance of 0.5 Da, enzyme trypsin with 2 missed cleavages, variable modification oxidation (M), fixed modification carbamidomethyl (C) and FDR targets 0.01 (strict) and 0.05 (relaxed).

The workflow included the Minora Feature Detector with min. trace length 7, S/N 3, PSM confidence high. Percolator was used to filter for highly confident targets. The consensus workflow in the PD software was used to evaluate the peptide identifications and to measure the abundances of the peptides based on the LC-peak intensities, with Top3 abundance calculation, peptide-based imputation, low abundance resampling for imputation and protein-based ratio calculation. The *Arabidopsis thaliana* TAIR10 was used as protein database. The results were exported to an excel spreadsheet (Supplementary Data 2, 3). Gene ontology search was performed using Panther 19.0 (https://pantherdb.org/) using proteins with an enrichment (IP in VRN1-FLAG/ IP in WT FRI) higher than 2-fold.

### Fluorescence recovery after photobleaching (FRAP)

FRAP was performed according to the procedure previously described[87]. For H2A.Z, pHTA9::HTA9-GFP[34] FRI and pHTA9::HTA9-GFP *vrn1-4* FRI were used. For H3.3, pHTR5:HTR5::GFP[40] FRI and pHTR5:HTR5::GFP *vrn1-4* FRI were used. Seven-day-old seedlings grown at warm temperature were used for microscopy. Images were acquired with a Zeiss 780 confocal microscope and processed using Fiji. All quantitative values represent the mean of 5 (for HTA9-GFP) or 10 cells (for HTR5-GFP). The average recovery curves were used to fit a single exponential function and determine mobile fraction and average half-time recovery.

### Chromatin extraction under low salt

Chromatin was digested and extracted under low salt based on a procedure described in ref. 42. Four grams of plant material was ground in ice-cold nuclei isolation buffer (10 mM MES-KOH pH 5.4; 10 mM NaCl; 10 mM KCl; 2.5 mM EDTA; 250 mM sucrose; 0.5 mM spermidine; 1 mM DTT; 1X cOmplete protease inhibitors), until the tissue was completely homogenised. The homogenate was filtered through two layers of Miracloth and Triton X-100 was added to a final concentration of 0.4%. The homogenate was centrifuged at $1000 \times g$ for 10 min at 4 °C. Nuclei were washed twice with MNase digestion buffer (10 mM Tris-HCl pH 7.5; 5 mM NaCl; 2.5 mM CaCl₂; 2 mM MgCl₂; 1X cOmplete protease inhibitors) and digested with 10 U/ml MNase (TaKaRa) for 10 min at 37 °C. The reaction was stopped by addition of EGTA to 2 mM. Digested chromatin was solubilized with either 80 mM Triton buffer (70 mM NaCl; 10 mM Tris-HCl pH7.5; 2 mM MgCl₂; 2 mM EGTA; 0.1% Triton X-100; 1X cOmplete protease inhibitors) for 15 min or 600 mM Triton buffer (585 mM NaCl; 10 mM Tris-HCl pH 7.5; 2 mM MgCl₂; 2 mM EGTA; 0.1% Triton X-100; 1X cOmplete protease inhibitors) for 2 h. The supernatant from each extraction was treated with 5 µg of RNase A at 37 °C for 30 min. SDS was added to a final concentration of 0.63%, and samples were treated with 50 µg proteinase K at 55 °C for 1 h. DNA was extracted with phenol-chloroform and used for qPCR analysis. qPCR data was normalised to undigested genomic DNA and *ACTIN2*. Primers are listed in Supplementary Data 1.

### MNase-seq

WT FRI and *vrn1* FRI seedlings were grown at warm temperature and subject to 1 week of cold. Three biological replicates were used for warm temperature and four for 1 week of cold. The results for each replicate are shown (Supp Fig. 4). 1.5 g of plant material per replicate was ground with liquid nitrogen to a fine powder using a pestle and mortar. The tissue was resuspended in 35 mL of Nuclei Isolation Buffer (20 mM HEPES-KOH pH 7.4; 0.44 M sucrose; 1.25% ficoll; 2.5% Dextran T40; 20 mM KCl; 0.5 mM spermidine; 5 mM DTT; 1% Triton X-100; 1X cOmplete protease inhibitors; and 0.1 mM PMSF). The homogenate was filtered through two layers of Miracloth, and centrifuged at $2000 \times g$ for 15 min at 4 °C. The nuclei were washed once with 5 mL of Nuclei Isolation Buffer and purified on a Percoll gradient as following: 2 mL of 75% Percoll (Merck, P7828) in Nuclei Isolation Buffer topped with 2 mL of 40% Percoll in Nuclei Isolation Buffer topped with 2 mL of

nuclei resuspended in Nuclei Isolation Buffer in a 15 mL tube. Purified nuclei were obtained in between the layers containing 40% and 75% Percoll after centrifuging at 7000 $g$ for 30 min at 4 °C and washed once more in 6 mL Nuclei Isolation Buffer. The nuclei were resuspended in 1.5 mL MNase Buffer (20 mM Tris pH 8.0; 0.125 M sucrose; 5 mM NaCl; 40 mM KCl; 2 mM CaCl$_2$; 1X cOmplete protease inhibitors; and 0.1 mM PMSF), transferred to a clean Eppendorf tube, and centrifuged at 1500 × $g$ for 10 min at 4 °C. After centrifuging, the buffer was completely removed, and the nuclei pellet weight was determined by subtracting the weight of the empty tube using an analytical balance scale. 65 mg of purified nuclei in 300 uL of MNase Buffer were treated with 10 U of MNase (Worthington) for 30 min at 37 °C. Reactions were quickly stopped by adding 16 μL of 0.5 M EDTA, 41 μL of 5 M NaCl, 40 μL of 10% SDS, and 3 μL of 20 mg/ml Proteinase K. Samples were incubated at 65 °C for 1 h, and DNA was extracted with two rounds of phenol-chloroform pH 8, one round of chloroform, and precipitation with 15 μg glycoblue and 2.5 volumes of cold pure ethanol, at −80 °C for 1 h. DNA was pelleted by centrifugation, washed once with 75% ethanol, and resuspend in 30 μL of H$_2$O. DNA samples were incubated with 5 μg of RNase A for 20 min at 37 °C. Mono- and sub-nucleosomal DNA fragments were purified with AMPure XP beads (Beckman Coulter), recovering all fragments <200 bp and excluding all fragments >300 bp.

Sequencing libraries were prepared using the NEBNext Ultra II DNA Library Prep Kit (NEB, #E7645) following the manufacturer's instructions. After adaptor ligation, DNA was purified with AMPure XP beads, recovering all fragments >200 bp. Illumina indexing PCR was performed with 8 cycles. Libraries were pooled and enriched for target loci in one reaction using myBaits Custom Kit (Daicel Arbour Biosciences) following the manufacturer's instructions for high-sensitivity hybridisation capture for targeted NGS. Table 1 lists the target genomic loci for which baits were requested from Daicel Arbour Biosciences. The bait probes consist of 80 nt-long oligos that overlap by 40 nt spanning the entire target region. The sequence of all baits is listed in Supplementary Data 4. The bait-enriched libraries were sequenced with Illumina HiSeq Xten PE150. At least three biological replicates were sequenced ($n = 3$ for WT FRI warm and $vrn1$-$4$ FRI warm, and $n = 4$ for WT FRI cold and $vrn1$-$4$ FRI cold). Data for each individual replicate is shown in Supplementary Fig. 6.

### MNase-seq data analyses
Demultiplexed FASTQ data was obtained from BGI Tech Solutions. Illumina adaptor sequences were trimmed with fastp[88]. Paired-end reads were aligned to the Arabidopsis TAIR10 reference genome using Bowtie2 v2.4.1[89] with the parameters --end-to-end --no-unal. Samtools v1.5[90] was used to convert the SAM files into BAM files and sort reads by genomic coordinate, and then to split the sorted BAM files into files containing only reads mapped to each bait-target locus listed in Table 1. At this step, reads with a base quality under 10 were discarded. Further analyses were performed for each locus individually using in-house scripts (https://github.com/Miguel-Montez/Targeted-MNase-seq-analysis) in RStudio v4.2.3. BAM files were imported to RStudio and converted into a dataframe with the following columns: read pair id, insert leftmost coordinate, and insert length. That information was used to compute the insert rightmost coordinate (insert leftmost coordinate + insert length). Nucleosomal fragments were then reconstituted for each read pair based on leftmost and rightmost coordinates. MNase-seq plots show each nucleosomal DNA fragment as a horizontal coloured segment along the sequence on the $x$-axis. The length of the fragment was resolved on the $y$-axis. The colour and thickness of the segment indicate the average frequency among the biological replicates, where frequency is the number of DNA fragments with the same start and end positions and same size, divided by the total number of fragments over the locus. For analyses of nucleosomal and subnucleosomal fragments, only lengths between 100 bp and

150 bp were kept. For analyses of fragments potentially reflecting adjacent di-nucleosomes, only lengths between 230 bp and 300 bp were kept. Based on the alternative positions for the +1 nucleosome that were observed on the MNase plots, we defined two windows that would encompass the majority of the +1 nucleosomal DNA fragments at a TSS-proximal position and at a TSS-distal position. Cold-induced nucleosome repositioning was assessed by quantifying the change in the ratio of TSS-proximal to TSS-distal fragments in the different genotypes in the cold relative to the warm. For analysis of centre position, 20,000 nucleosomal fragments were randomly pooled from each sample per genotype per treatment. The centre position was plotted as a function of sequence over 250 bp upstream and downstream *FLC* TSS. For analysis of relative nucleosome occupancy at *FLC*, we started by defining the windows encompassing the majority of fragments that could correspond to −1, +1, +2 and TTS-proximal nucleosomes. For each of such nucleosome positions, we normalised the number of nucleosomal fragments to the number of fragments on two control loci (*ACT7* and *GAPDH*) in each library aiming to reduce the sequencing depth impact.

### In silico nucleosome prediction
Prediction of nucleosome occupancy at the *FLC* locus was performed based on its genomic DNA sequence through the web server http://bio.physics.leidenuniv.nl/~noort/cgi-bin/nup3_st.py[91].

### MD simulations
The *FLC* locus was modelled using our chemically-specific coarse-grained model of chromatin[13]. In our model, nucleosomes are described at a one bead per amino acid resolution, where beads corresponding to charged amino acids, namely, lysine, arginine, aspartic acid, glutamic acid and histidine carry the total charge of their atomistic counterpart at pH-7. All amino acid beads have a relative hydrophobicity and diameter defined from atomistic simulations and experimentally measured data[13]. The core histone protein is coupled to an elastic network model (ENM), that enforces secondary structure based on the crystal structure of each respective histone, while histone tails are treated as fully flexible polymers. DNA is represented in a Rigid Base Pair (RBP) model, that represents every single base pair as an ellipsoid with added phosphate charges. The sequence-dependent behaviour of the DNA is captured by parametrising the RBP with DNA mechanical potential energies derived from MD atomistic simulations[92,93]. All model parameters and energy functions can be found in ref. 13.

Nucleosomes were positioned according to the most probable positions from our MNase-seq. For the "warm-positioned" models, the most probable positions before cold were used, with nucleosomes initially spaced by 143, 1, and 21 bp. An alternative warm configuration of 3 nucleosomes spaced by 260 and 52 bp was also generated. For the "cold-positioned" model, nucleosomes were placed according to the most probable positions in the cold, with nucleosomes spaced by 92, 34 and 39 bp. In the H2A.Z simulations, the two copies of histone H2A were substituted by histone variant H2A.Z. For the simulations with "rigid" nucleosomes, an elastic network model was added between histone core and DNA to constrain the breathing and sliding of the nucleosomes.

Simulations were performed using LAMMPS[94] (version 3rd March 2020) with our custom code[13]. We performed Debye-Hückel replica exchange, with 16 replicas for each simulation condition. The simulations were run at a constant temperature of 300 K using the Langevin thermostat. The Debye-Hückel screening parameter κ ranged from 8 Å$^{-1}$ to 15 Å$^{-1}$, with spacings optimised to ensure an exchange rate near 0.3. encompassing physiologically relevant salt concentrations. Each replica was simulated at a 10 fs timestep, for 3 μs, with configurations saved every 10000 fs for analysis. The first 500 ns were discarded from the simulation before analysis. Contact analysis and the analysis of

**Table 1 | List of target genomic loci for bait enrichment**

| chromosome | start_coordinate | end_coordinate | strand | locus_identifier | gene_name |
|---|---|---|---|---|---|
| 1 | 2256384 | 2261187 | – | AT1G07350 | SR45A |
| 1 | 2460790 | 2465729 | – | AT1G07940 | EF-1a |
| 1 | 4537668 | 4544889 | + | AT1G13260 | RAV1 |
| 1 | 4563160 | 4567935 | – | AT1G13320 | PP2A |
| 1 | 4607077 | 4612731 | – | AT1G13440 | GAPC-2 |
| 1 | 5674495 | 5679819 | – | AT1G16610 | SR45 |
| 1 | 16127298 | 16133286 | + | AT1G42970 | GAPDH |
| 1 | 22019377 | 22023503 | – | AT1G59830 | PP2A-1 |
| 1 | 23057582 | 23066563 | – | AT1G62360 | STM |
| 1 | 24326928 | 24338435 | + | AT1G65480 | FT |
| 2 | 7804371 | 7812380 | – | AT2G17950 | WUS |
| 2 | 12136241 | 12147883 | – | AT2G28390 | SAND |
| 2 | 12643917 | 12646560 | – | AT2G29550 | TUB7 |
| 2 | 17709441 | 17713196 | – | AT2G42540 | COR15A |
| 3 | 4591541 | 4595767 | – | AT3G13920 | EIF-4a |
| 3 | 8938949 | 8945385 | + | AT3G24520 | HSFC1 |
| 4 | 2716012 | 2720493 | + | AT4G05320 | UBQ10 |
| 4 | 8816815 | 8821651 | + | AT4G15415 | PP2A |
| 4 | 9205597 | 9215680 | – | AT4G16280 | FCA |
| 4 | 13014085 | 13024498 | – | AT4G25480 | DREB1ABC |
| 4 | 16070740 | 16075994 | + | AT4G33380 | AT4G33380 |
| 4 | 16402844 | 16407303 | – | AT4G34270 | TIP41 |
| 5 | 3047753 | 3055618 | + | AT5G09810 | ACT7 |
| 5 | 3172382 | 3181949 | – | AT5G10140 | FLC |
| 5 | 3959938 | 3966321 | – | AT5G12250 | TUB6 |
| 5 | 8967640 | 8970349 | + | AT5G25760 | UBC21 |
| 5 | 17683446 | 17686534 | + | AT5G43940 | ADH2 |
| 5 | 17800737 | 17806397 | – | AT5G44200 | CBP20 |
| 5 | 19178881 | 19183523 | + | AT5G47230 | ERF5 |
| 5 | 21239531 | 21244723 | + | AT5G52310 | RD29A |
| 5 | 23245395 | 23252989 | – | AT5G57380 | VIN3 |
| 5 | 24287596 | 24293336 | – | AT5G60390 | EIF1A |

inter-nucleosomal distances were done using the Visual Molecular Dynamics (VMD) software[95]. Radius of Gyration analysis and image rendering were done using the Open Visualisation Tool (OVITO) software[96]. For the Radius of Gyration analysis, since the values non-normally distributed, we calculated the median of the radius-of-gyration values and estimated its 95% confidence interval (CI) via bootstrapping with 10,000 resamples. Specifically, we repeatedly sampled the data *with replacement*, computed the median of each bootstrap sample, and took the 2.5th and 97.5th percentiles of the resulting bootstrap distribution as the lower and upper CI bounds. The median of the original dataset served as the central estimate.

### Statistical analysis
All statistical analyses were performed using RStudio v4.2.3. Sample sizes are mentioned in figure legends and/or displayed in plots as individual data points. Sample sizes were not pre-specified. Information regarding the statistical test applied to each data is mentioned in the corresponding figure legend. Significant differences were accepted at $p$-values < 0.05.

### Reporting summary
Further information on research design is available in the Nature Portfolio Reporting Summary linked to this article.

### Data availability
The MNase-seq data generated in this study have been deposited in the NCBI Gene Expression Omnibus (GEO) under the accession GSE274066. The mass spectrometry proteomics data have been deposited in the ProteomeXchange Consortium via the PRIDE[97] partner repository with the dataset identifiers PXD054597 (VRN1-FLAG IP-MS in the warm) and PXD060769 (VRN1-FLAG IP-MS during cold).

### Code availability
All the codes for MNase-seq data processing, analysis, and plotting are available on GitHub https://github.com/Miguel-Montez/Targeted-MNase-seq-analysis. All the custom code for the chromatin model can be found at https://github.com/CollepardoLab/CollepardoLab_Chromatin_Model.

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

## Acknowledgements

We thank Sheila Teves (University of British Columbia) for the assistance with low-salt chromatin extraction; Steven Henikoff (Fred Hutchinson Cancer Research Centre) for discussions on the MNase-seq data interpretation; Carlo Martins and Gerhard Saalbach (Proteomics Facility at John Innes Centre) for the mass spectrometry analysis service; Jitender Cheema (EMBL-EBI) for the help with processing raw sequencing data; Frédéric Berger (Gregor Mendel Institute, Vienna) for sharing the HTR5-GFP lines; and Richard Amasino for sharing the *vrn1* fast neutron mutant. We also thank two previous lab members Clare Lister and Joshua Mylne for their assistance with the genomic VRN1 cloning and allelism tests respectively. This work was funded by grants from the European Research Council (ERC) under the European Union's Horizon 2020 research and innovation programme; EPISWITCH-833254 to CD; and 803326 to RCG. CD was also funded by Wellcome Trust grant 210654/Z/18/Z and a Royal Society Professorship RP\R1\180002. RCG and MJM are also supported by the UK Research and Innovation (UKRI) Engineering and Physical Sciences Research Council (EPSRC) under the UK Government's guarantee scheme (grant EP/Z002028/1 awarded to RCG). MJM would like to acknowledge the Winton Programme for Physics of Sustainability for doctoral funding. JH is supported by the Herchel Smith Postdoctoral Fellowship Fund, and the EPSRC [grant number EP/X02332X/1] under the UKRI Postdoctoral Fellowships Guarantee Scheme [project TF-CHROM-LLPS]. This project made use of time on HPC granted via the UK High-End Computing Consortium for Biomolecular Simulation, HECBioSim (http://hecbiosim.ac.uk), supported by EPSRC (grant no. EP/X035603/1) to RCG.

## Author contributions

M.M., D.Z., B.R. and M.N. performed the experiments. J.H. and M.J.M. carried out the computational simulations. R.C.G. supervised the computational simulations. C.D. supervised the work. M.M., J.H., M.J.M., R.C.G. and C.D. wrote the manuscript.

## Competing interests

The authors declare no competing interests.
