## [Transparent Peer Review file · Nature Communications]

Cold-induced nucleosome dynamics linked to silencing of Arabidopsis *FLC*

Corresponding Author: Professor Caroline Dean

Version 0:

Reviewer comments:

Reviewer #1

(Remarks to the Author)

This paper investigated local changes in the chromatin state of Flowering Locus C (FLC) of Arabidopsis in cold environment, relative to warm condition, by various experiments complemented with computer simulations. Loss of VRN1 increased the H2A.Z levels right upstream of the FLC TSS, which enhanced nucleosome dynamics in the locus. Authors further suggest that repositioning of +1 nucleosomes in FLC locus happened and is coupled with transcription repression. The findings are sound, but I have a few concerns largely on the strength of the data.

1) A major finding is repositioning of +1 nucleosome in FLC in cold condition, and its reduction in the *vrn1* mutation (Fig. 3DE). The way the data was presented in Fig.3D does not look optimal to judge if the repositioning is significant or not, I consider. I would like to see a more straightforward "average frequency of DNA fragments as a function of sequence". Figure 3E is clear in the presentation, where, however, I do not see a significant difference between WT and the *vrn1* mutant. Thus, the data is not strong enough to support the authors' conclusion.

1b) A related minor point on Fig.3CD is that I did not well understand the meaning of the color/tile-size described in the right side. Possibly, this confused me in the above interpretation.

1c) For the connection to computer simulations, I would like to see some quantitative data on the -1 nucleosome position. The results of MD simulations are, in my understanding, largely derived from the position of +1 and -1 nucleosomes, but the experimental data for -1 nucleosome is currently vague.

2) As the function of VRN1, the recruitment of protein factors, the reduction in the levels of H2A.Z, and repositioning of +1 nucleosome are described. Their mutual couplings and their individual impacts are unclear to me. Especially, repositioning can simply be a by-product of factor-recruitments.

2b) It is reasonable to assume that VRN1 binding/PRC2 formation would change the chromatin structure and nucleosome dynamics. These effects are ignored in MD simulations: MD simulations focus on the inherent nucleosome dynamics. Authors need to discuss these points.

Minor points.

3) MD simulation methods need to be provided more comprehensively. For example, the number of replicas, the range of Debye-Huckel parameter in the replicas, the simulation length, the simulation temperature, the number of replicates of simulations must be given.

4) Figure 5C needs the estimate of standard error.

5) SuppFig6: Labels in the figure and in the caption are not consistent. Need renumbering.

Reviewer #2

(Remarks to the Author)

The manuscript investigates the mechanisms by which temperature and VRN1 influence nucleosome dynamics and gene expression. The authors utilize a broad range of experimental approaches to examine changes in nucleosome modification, positioning, H2A.Z levels, and protein-protein interactions. These experiments are complemented by computational modeling of chromatin organization, incorporating nucleosome positions identified through MNase-seq. The study presents an intriguing mechanism, which I find compelling. My review focuses on the computational aspects, and I offer the following comments for consideration.

First, the statement that “nucleosome breathing and sliding motions were allowed as dictated by the sequence-dependent mechanical properties” appears to be imprecise. While I understand that the model permits nucleosome sliding along the DNA if sufficient simulation time is provided, the results in Figure S5 suggest that the simulations do not extend to the sliding timescale. In other words, no significant sliding can be observed along the simulations. However, I am open to being corrected on this point.

In Figure 4, the authors propose a model describing the influence of temperature on nucleosome dynamics, suggesting that reduced temperature repositions the +1 nucleosome closer to the transcription start site (TSS). The authors indicate this repositioning might be driven by chromatin remodelers. It's unclear why the remodeler only gets activated at low temperatures.

Could this shift simply reflect a thermodynamic preference, where binding at the new position is energetically more favorable? If the authors can explore nucleosome sliding using their near-atomistic model, I suggest investigating the free energy difference between the two positions.

My understanding of VRN1 is that it hinders H2A.Z accumulation, leading to slow transitions between states and resulting in well-positioned nucleosomes. In the mutant, H2A.Z promotes nucleosome instability, accelerating transitions and yielding less defined positions. Is this interpretation accurate? If so, the modeling in Figure S7 seems less relevant to the experimental observations. Instead, the role of H2A.Z may involve modulating the distribution of nucleosome positions, suggesting the chromatin structure is a mixture of the two ensembles shown in Figure 5B.

Lastly, I find the following statement unclear:

“This suggests that the cold-induced nucleosome repositioning may switch the local chromatin from a transcription-accessible to a Polycomb nucleation-accessible state.” Are the authors proposing that the conformational changes observed in simulations correspond to transcription-accessible (warm) and Polycomb nucleation-accessible (cold) states? If so, why would a more compact +1/+2/+3 structure be more transcriptionally accessible? Clarifying this point would greatly enhance the reader's understanding.

Reviewer #3

(Remarks to the Author)

This study focuses on how VRN1, a non-sequence-specific DNA binding protein regulates FLC chromatin and expression under prolonged cold treatment. In *vrn1* mutant, the repressive histone marks H3K27me3 and H2Aub cannot be accumulated, while H2A.Z shows increased accumulation levels, revealing that VRN1 can affect FLC chromatin through modulating histone modifications and variants. Moreover, the study shows that both cold treatment and VRN1 can contribute to chromatin dynamics. Although the model shown in the study will potentially help understand the mechanisms by which how VRN1 and cold regulate FLC chromatin during vernalization, the evidences shown in the manuscript need to be confirmed by additional experiments.

1. IP-MS was conducted to detect proteins associated with VRN1-FLAG from crosslinked seedlings. Although a large number of proteins associated with chromatin were identified as VRN1-interacting proteins, the possible contamination cannot be excluded. Plants without the VRN1-FLAG transgene should be used as a negative control in IP-MS. Moreover, the interactions should be validated at least for part of them.
2. MNase-seq plots were shown in Figure 3C and 3D. However, these figures did not show any visible changes between different samples. A statistical analysis was performed in Figure 3E. However, it is weird that there are three values for the first three samples but four values for the last sample (*vrn1* FRI 1W cold). The variant of the last sample is clearly higher than those of the three other ones. All the four values should be included in the statistical analysis.
3. FRAP was used for testing the impact of *vrn1* and cold on chromatin dynamics, indicating that both H2A.Z-containing chromatin and H3.3-containing chromatin are more dynamic in *vrn1* than in WT. Because the *vrn1* mutant specifically showed flowering time alteration but had no other effect. It is difficult to understand why the *vrn1* mutation has significant effect on chromatin dynamics but does not affect genome-wide gene expression or development. Moreover, an explanation should be provided to show why H2A.Z and H3.3 but not canonical H2A or H3 were used in the analysis. At least, the canonical histones should be used as a negative control.
4. The nucleosome position was thought to be affected by cold and by *vrn1* mutation. However, the results shown in Figure 3D and 3E seem not convincing.

5. In Figure 3A, five loci were selected for the detection of solubilised chromatin. Although the effect of cold treatment and *vrn1* mutation on H3K27me3 and H2Aub were detected specifically in the regions nearby the transcription start site and intragenic regions (Figure 1), the effect of cold treatment and *vrn1* mutation on the solubilised chromatin did not show any locus specificities. Especially, the cold treatment significantly reduced the solubilised chromatin level at all the five tested loci (Figure 3A). I wonder whether the experiment can really detect the chromatin state *in vivo*.

6. Figure 4 shows a model for the proposed effect of nucleosome dynamics on chromatin structure change at FLC locus, while Figure 5 illustrates the simulation of FLC chromatin structure based on the DNA sequence and nucleosome position. The model and the simulations are totally based on the results shown in Figure 3. As indicated above, the results shown in Figure 3 need to be further confirmed using additional data.

7. In line 235, "VRN1 genetically depends on ARP6 to silence FLC, implicating H2AZ eviction as a functional requirement for FLC silencing in the cold". This statement is not correct. ARP6, as a SWR1 complex, promotes the deposition of H2A.Z on chromatin. The ARP6-dependent deposition of H2A.Z on the FLC locus enhances FLC expression but not for silencing FLC. In fact, VRN1 and ARP6 have opposing effects on FLC expression.

Version 1:

Reviewer comments:

Reviewer #1

(Remarks to the Author)

Authors largely revised the manuscript according to reviewers' comments. I recognized that all of my points/concerns were addressed appropriately.

Reviewer #2

(Remarks to the Author)

The authors have successfully addressed my comments and I support its publication.

Reviewer #3

(Remarks to the Author)

My concerns on the previous version have been well addressed in the revised manuscript, and I have no additional comments.

made.

Dear Editor,

We are pleased to provide a revised version of our manuscript. We sincerely appreciate the time and effort the Reviewers dedicated to understanding our work and providing constructive feedback. As is often the case, their reviews have significantly helped improve the manuscript. The Reviewers raised several important questions and provided stimulating discussion points, which we have addressed. We are happy for the peer review file to be made public in full upon publication. A major concern shared by two Reviewers was the way the MNase-seq data was presented and whether the differences observed supported our conclusions. In response, we have substantially revised the manuscript to address this and all other major and minor comments. We have also added new analyses of the data, and new experimental/computational results, including a new IP-MS dataset (deposited to the PRIDE repository under the identifier PXD060769), ChIP-qPCR, and we have reproduced all the molecular dynamics (MD) simulations over three times their previous duration. We have added and modified many figure panels. We believe this has strengthened support for our conclusions and made several points clearer to the reader. Below, we provide a point-by-point response. We also provide a manuscript text with track changes where all the revisions are highlighted, and references to specific lines are made in our response.

Reviewer #1 (Remarks to the Author)

This paper investigated local changes in the chromatin state of Flowering Locus C (FLC) of Arabidopsis in cold environment, relative to warm condition, by various experiments complemented with computer simulations. Loss of VRN1 increased the H2A.Z levels right upstream of the FLC TSS, which enhanced nucleosome dynamics in the locus. Authors further suggest that repositioning of +1 nucleosomes in FLC locus happened and is coupled with transcription repression. The findings are sound, but I have a few concerns largely on the strength of the data.

1) A major finding is repositioning of +1 nucleosome in FLC in cold condition, and its reduction in the *vrn1* mutation (Fig. 3DE). The way the data was presented in Fig.3D does not look optimal to judge if the repositioning is significant or not, I consider. I would like to see a more straightforward “average frequency of DNA fragments as a function of sequence”. Figure 3E is clear in the presentation, where, however, I do not see a significant difference between WT and the *vrn1* mutant. Thus, the data is not strong enough to support the authors’ conclusion.

The Reviewer asks for a more straightforward plot showing the nucleosome fragments as a function of sequence. We agree that the original plots are not intuitive and require a lot of effort to be understood.

We have further analysed the MNase-seq and redrawn the plots replacing **Fig3D** in the new version of the manuscript. The **new Fig3D** shows the nucleosome’s centre positions as a function of sequence along a window of 250 bp downstream and upstream of *FLC* TSS. This plot provides a high-resolution, view of the +1 and -1 nucleosome positions. The +1 repositioning is now visually much clearer – while a predominant peak for the +1 centre position is observed in WT in the warm, in the cold, a TSS-proximal double-peak is notable and represents the cold-induced +1 repositioning. It is also visually clearer that this cold-induced repositioning is affected in the *vrn1* mutant.

The Reviewer also raised concerns about the changes in the mutant. We provide an alternative plot in the **new Fig3E** with a better representation of the cold-induced repositioning by showing the ratio of TSS-proximal +1 (repositioned) to TSS-distal +1 (not repositioned) in the cold normalised to the warm. We have also tested for significant differences. While the cold-induced repositioning is statistically significant in the WT, in the *vrn1* mutant, it is not.

We also provide new plots in **SuppFig5A, B**. These simply display the mean nucleosome centre position (+/- standard deviation). This offers an alternative quantification of nucleosome repositioning without normalisation. The results in **SuppFig5A** show that the +1 nucleosome's centre position is significantly closer to the TSS after cold in the WT but not in the mutant. It also shows a significant difference in the mean centre position between WT and *vrn1* in the cold. We believe these new plots make the conclusions clearer, provide the readers with a more straightforward presentation, and thus improve the manuscript.

The original Fig3D has been moved to the **new SuppFig5C**. This is because even though we agree the plots are less intuitive to understand, they are more informative. These show the nucleosome positions by colouring the entire length of the nucleosomal DNA fragments protected from MNase digestion. The plots also provide a third dimension to the positioning as a function of sequence, which is given by the length distribution (from 100 to 150 bp) on the y-axis. This makes visual comparisons harder but allows the reader to see the entire nucleosomal DNA from the leftmost to rightmost end.

1b) A related minor point on Fig.3CD is that I did not well understand the meaning of the color/tile-size described in the right side. Possibly, this confused me in the above interpretation.

We have updated the methods section to include a better description of the analysis and plots. Lines 641-645 of the text with track changes. "MNase-seq plots show each nucleosomal DNA fragment as a horizontal-coloured segment along the sequence on the x-axis. The length of the fragment was resolved on the y-axis. The colour and thickness of the segment indicate the average frequency among the biological replicates, where frequency is the number of DNA fragments with the same start and end positions and same size, divided by the total number of fragments over the locus".

1c) For the connection to computer simulations, I would like to see some quantitative data on the -1 nucleosome position. The results of MD simulations are, in my understanding, largely derived from the position of +1 and -1 nucleosomes, but the experimental data for -1 nucleosome is currently vague.

The Reviewer asks for a quantification of the -1 nucleosome position. The new plots in **Fig3D** and **SuppFig5A, B** quantify the positioning of the -1 nucleosome side by side with the +1 nucleosome. The results indicate that the -1 nucleosome has a predominant and well-defined position, which is less variable across treatments and genotypes compared to the +1 nucleosome.

2) As the function of VRN1, the recruitment of protein factors, the reduction in the levels of H2A.Z, and repositioning of +1 nucleosome are described. Their mutual couplings and their individual impacts are unclear to me. Especially, repositioning can simply be a by-product of factor-recruitments.

We agree that the previous version of the text did not clearly explain the relationship between the different aspects characterised. The manuscript text has been substantially revised, and more effort has been put into integrating the multiple aspects of *FLC* regulation. Briefly, we propose that cold alters the local nucleosome dynamics and 3D structure of the chromatin at *FLC*, mediating a transition from a state prone to transcription to a state that can be silenced by Polycomb. VRN1 plays an important role in this cold-induced process by limiting H2A.Z incorporation. The *vrn1* mutation causes a hyper-accumulation of H2A.Z at the first nucleosomes within the nucleation region of *FLC*, affecting the local nucleosome dynamics and structure of the chromatin in response to cold. The simulations of *FLC* chromatin with nucleosomes initially positioned as most frequently observed in the cold show that H2A.Z incorporation alone (without any other changes or activity/recruitment of protein factors) induces nucleosome dynamics and promotes a more compact local chromatin structure. This is more similar to the chromatin in the warm than to that in the cold. These results suggest that the effects of *vrn1* mutation on *FLC* chromatin may be a consequence of high H2A.Z levels. The impact of VRN1 on H2A.Z is supported by its interaction with INO80 *in vivo*, and the genetic epistasis between *vrn1* and *arp6*.

The document with track changes shows the edits that clarify the relationship between the different components. Lines 98-103, the link between *vrn1* mutation and H2A.Z supported by genetic analysis. Lines 153-157, the link between H2A.Z accumulation and high nucleosome dynamics at *FLC*. Lines 267-272, the link between the local chromatin structure in the warm and a high transcriptional state. Lines 277-286, the link between the cold-induced +1 repositioning, local structure changes, and the transition to a state that is accessible to PRC2 silencing. The model described in lines 224-228 summarises many of these aspects.

The Reviewer also asks whether the nucleosome repositioning could be a consequence of the recruitment of protein factors. We discuss the potential components involved in the +1 repositioning, including antisense transcription, histone deacetylation, and the impact of different proteins in lines 391-398 and 427-439. VRN1 associates with *FLC* around the TSS and with many protein factors. Although VRN1-chromatin association could potentially impact nucleosome positioning, we do not see strong evidence for that. VRN1-chromatin association is detected both in the warm and cold at similar levels, and no major changes are observed at the -1 nucleosome where the VRN1 ChIP signal is found. Other proteins also associate with *FLC* chromatin both in the warm and cold. PRC2 and PRC2-accessory proteins such as VIN3 are targeted to the *FLC* nucleation region in the cold. VIN3 contains a protein polymerisation domain involved in promoting higher-order assemblies to enhance chromatin association and H3K27me3 nucleation (Schulten *et al.*, 2024 bioRxiv doi: 10.1101/2024.02.15.580496). It is an interesting possibility that this protein would be involved in the +1 nucleosome repositioning, even though it is not known to interact with any chromatin remodelling complexes. This has been mentioned in the Discussion section.

2b) It is reasonable to assume that VRN1 binding/PRC2 formation would change the chromatin structure and nucleosome dynamics. These effects are ignored in MD simulations: MD simulations focus on the inherent nucleosome dynamics. Authors need to discuss these points.

As discussed above, the repositioning could indeed be promoted by the association of specific proteins with the chromatin. We agree with the Reviewer that exploring the impacts of proteins known to

interact with *FLC* chromatin at the nucleation region would help us understand their impact on nucleosome repositioning and local chromatin structure changes. However, further MD simulations with *FLC* chromatin and candidate proteins and protein complexes extend beyond the current work and will require substantial new computational simulations as well as additional experimental data, to assess the individual impact of the proteins in question. The additional simulations are far from trivial as they would require a multiscale approach including large-scale atomistic simulations to ensure our model correctly balances the relative binding affinities of the candidate proteins to DNA and histones, while preserving the intrinsic binding energy that maintains DNA wrapping around the nucleosome. In this work, we aimed to simulate the chromatin structure dynamics linked to the cold-induced nucleosome repositioning and a state that would explain the observed impact of VRN1. This was achieved via H2A.Z variant incorporation. Importantly, the fact that the simulations focus on the inherent nucleosome dynamics without protein factors shows that nucleosome repositioning and H2A.Z alone alter the local chromatin structure to one consistent with *FLC* repression.

Minor points.

3) MD simulation methods need to be provided more comprehensively. For example, the number of replicas, the range of Debye-Huckel parameter in the replicas, the simulation length, the simulation temperature, the number of replicates of simulations must be given.

We thank the reviewer for raising our attention to the comprehensive description of the methods. The new version of the manuscript text has been extensively modified and now provides a much more detailed description of the MD simulations. See lines 667-706.

4) Figure 5C needs the estimate of standard error.

Following the Reviewer's suggestion, an estimate of error has been added to the plot in **Fig5C** and all other distribution plots from the MD simulations. For that, we added a vertical line indicating the median value with a shaded area for a 95% confidence interval.

5) SuppFig6: Labels in the figure and in the caption are not consistent. Need renumbering.

Thank you – the numbers in the figure legend had not been correctly assigned to the figure panels in the original SuppFig6, now **SuppFig9**. This has been corrected.

Reviewer #2 (Remarks to the Author):

The manuscript investigates the mechanisms by which temperature and VRN1 influence nucleosome dynamics and gene expression. The authors utilize a broad range of experimental approaches to examine changes in nucleosome modification, positioning, H2A.Z levels, and protein-protein interactions. These experiments are complemented by computational modeling of chromatin organization, incorporating nucleosome positions identified through MNase-seq. The study presents an intriguing mechanism, which I find compelling. My review focuses on the computational aspects, and I offer the following comments for consideration.

First, the statement that “nucleosome breathing and sliding motions were allowed as dictated by the sequence-dependent mechanical properties” appears to be imprecise. While I understand that the model permits nucleosome sliding along the DNA if sufficient simulation time is provided, the results in Figure S5 suggest that the simulations do not extend to the sliding timescale. In other words, no significant sliding can be observed along the simulations. However, I am open to being corrected on this point.

We thank the Reviewer for raising this important point. Indeed, the timescales of spontaneous nucleosome sliding are known to be very slow, typically occurring over minutes to hours in the absence of chromatin remodelers. As such, these are rare events that are not expected to occur spontaneously in standard, unbiased MD simulations, particularly on the microsecond timescales commonly accessible with coarse-grained methods. To overcome this limitation, we employed Hamiltonian Replica Exchange Molecular Dynamics (H-REMD) simulations, which enhance the sampling of rare events such as nucleosome sliding. Additionally, by using a coarse-grained model with implicit solvent, the high-frequency vibrations present in the system are averaged out, effectively accelerating the dynamics with respect to atomistic MD simulations. As a result, the characteristic timescales in our coarse-grained simulations are significantly faster than those of atomistic MD with explicit solvent, making it feasible to observe dynamic processes that would otherwise remain inaccessible within standard simulation windows. To ensure adequate sampling, we have extended all simulations to 3 μ s, for an accumulated sampling of 48 μ s per H-REMD set of simulations. We have added this information to the revised manuscript. These extended simulations revealed that while sliding is limited under cold conditions, it becomes more pronounced under warm conditions, and is further enhanced upon H2A.Z incorporation (see SuppFig8, 10).

In Figure 4, the authors propose a model describing the influence of temperature on nucleosome dynamics, suggesting that reduced temperature repositions the +1 nucleosome closer to the transcription start site (TSS). The authors indicate this repositioning might be driven by chromatin remodelers. It's unclear why the remodeler only gets activated at low temperatures.

We thank the reviewer for this point. We were not clear enough about our hypothesis involving chromatin remodelers in the previous version of the manuscript. Our data, including genetic analysis between *vrn1* and *arp6*, ChIP and FRAP for H2A.Z in *vrn1*, IP-MS for VRN1, and MD simulations with H2A.Z, suggest that VRN1 functions in limiting H2A.Z incorporation, having an impact on nucleosome dynamics and *FLC* silencing. The data also points to chromatin remodelers being involved in this process. This has been made clearer in the new version of the manuscript. However, we have no indications that the cold-induced repositioning of the *FLC* +1 nucleosome is directly mediated by chromatin remodelers. We hypothesise that chromatin remodelers such as the INO80 complex have an impact on the nucleosome dynamics via H2A.Z in a VRN1-dependent manner, but that is not cold-specific. The factors that mediate the repositioning in the cold are yet to be identified and may not be chromatin remodelers.

Could this shift simply reflect a thermodynamic preference, where binding at the new position is

energetically more favorable? If the authors can explore nucleosome sliding using their near-atomistic model, I suggest investigating the free energy difference between the two positions.

We appreciate the Reviewer's suggestion regarding quantifying the free energy difference between the two positions. One possible approach to estimate such free energy difference would involve constructing a Markov State Model (MSM) based on sufficient sampling of transitions between the two nucleosome positions, or alternatively, performing umbrella sampling simulations. However, such an analysis is not easily feasible in our current setup for several reasons. First, while umbrella sampling could yield the free energy difference between chromatin states with distinct nucleosome positions, identifying a suitable reaction coordinate for such a calculation is non-trivial, as the nucleosome configurations in the two states differ across the entire region—not just at the +1 nucleosome—making it challenging to define a continuous path between the two states. Second, an MSM would enable the reconstruction of the free energy landscape from the relative populations of metastable microstates, without the need for a reaction coordinate. However, our dataset does not capture enough transitions to build a statistically reliable MSM, and as noted above, the “warm” and “cold” chromatin systems we simulate are fundamentally different—not only in the position of the +1 nucleosome, but in the arrangement of all nucleosomes. To compare the structural dynamics and stability of the two chromatin states we treat them as separate systems, each constructed to reflect experimentally observed nucleosome configurations and histone variant distributions. A direct thermodynamic analysis of nucleosome repositioning within a unified energy landscape would require a different simulation design and significantly more sampling. This would constitute a substantial body of work beyond the scope of the present study.

My understanding of VRN1 is that it hinders H2A.Z accumulation, leading to slow transitions between states and resulting in well-positioned nucleosomes. In the mutant, H2A.Z promotes nucleosome instability, accelerating transitions and yielding less defined positions. Is this interpretation accurate? If so, the modeling in Figure S7 seems less relevant to the experimental observations. Instead, the role of H2A.Z may involve modulating the distribution of nucleosome positions, suggesting the chromatin structure is a mixture of the two ensembles shown in Figure 5B.

We thank the reviewer for this comment. Indeed, our hypothesis is that VRN1 limits H2A.Z accumulation, resulting in less dynamic nucleosomes. In that regard, the simulations in the old SuppFig7 (now SuppFig10) represent a system that mimics the *vrn1* mutant (with H2A.Z accumulation) and where we expect more mobile nucleosomes. Consistent with the experiments, in these additional simulations of H2A.Z-rich chromatin, we observe that lower nucleosome stability intrinsically emerges from replacing the canonical H2A histone with H2A.Z, and that subsequently gives rise to higher variations in chromatin configurations. This also increases chromatin compaction. We have revised the manuscript to include a discussion on the effect of the conformational variability induced by H2A.Z and its implications for chromatin structure under different conditions. We have clarified this connection in the text and ensured the relevance of the modelling to the experimental observations is better articulated.

Lastly, I find the following statement unclear: “This suggests that the cold-induced nucleosome repositioning may switch the local chromatin from a transcription-accessible to a Polycomb nucleation-

accessible state.” Are the authors proposing that the conformational changes observed in simulations correspond to transcription-accessible (warm) and Polycomb nucleation-accessible (cold) states? If so, why would a more compact +1/+2/+3 structure be more transcriptionally accessible? Clarifying this point would greatly enhance the reader's understanding.

Indeed, that is our hypothesis. The simulations predict that the 3D structure of *FLC* chromatin over the entire 5' end region is more open in plants in the warm than in the cold. In the warm, the -1 nucleosome remains distant from the rest, while the +1, +2 and +3 form a compact local structure. This promotes a larger and more accessible NDR, which is expected to stimulate transcription initiation. We speculate that the RNA polymerase II complex equipped with its various accessory proteins can similarly transcribe through these structures as it transcribes through a single nucleosome, especially when assisted by chromatin remodellers such as the FACT complex. This has been discussed in the text in lines 467-473. Adjacent +1 and +2 nucleosomes, as observed at *FLC* in the warm are also observed at promoter-proximal regions of actively transcribed human genes (see refs 69 and 70), suggesting these do not repress transcription and might even be promoted by it. However, the compact +1/+2/+3 structure might be a barrier to PRC2 activity. Given the available structures of PRC2 bound to nucleosomes, it seems unlikely that PRC2 can methylate H3K27 on the nucleosomes in the +1/+2/+3 compact structure. Nucleosome repositioning and local decompaction have been proposed to be involved in Polycomb silencing, as we discuss in the text, lines 401-411.

Reviewer #3 (Remarks to the Author)

This study focuses on how VRN1, a non-sequence-specific DNA binding protein regulates FLC chromatin and expression under prolonged cold treatment. In *vrn1* mutant, the repressive histone marks H3K27me3 and H2Aub cannot be accumulated, while H2A.Z shows increased accumulation levels, revealing that VRN1 can affect FLC chromatin through modulating histone modifications and variants. Moreover, the study shows that both cold treatment and VRN1 can contribute to chromatin dynamics. Although the model shown in the study will potentially help understand the mechanisms by which how VRN1 and cold regulate FLC chromatin during vernalization, the evidences shown in the manuscript need to be confirmed by additional experiments.

1. IP-MS was conducted to detect proteins associated with VRN1-FLAG from crosslinked seedlings. Although a large number of proteins associated with chromatin were identified as VRN1-interacting proteins, the possible contamination cannot be excluded. Plants without the VRN1-FLAG transgene should be used as a negative control in IP-MS. Moreover, the interactions should be validated at least for part of them.

The Reviewer is right to assume that contamination cannot be excluded if a control is not used. That is why we used WT FRI plants without the VRN1-FLAG transgene as a negative control in IP-MS. The results are analysed to find the enrichment ratios instead of the number of peptides. **Only proteins enriched in the VRN1-FLAG IP relative to WT IP were considered.** We have made this explicit in the Results section lines 117-118 (see version with track changes). This information is also in the legend of Fig2.

The Reviewer also asks for a validation for the interactions observed before (Fig2D). **We have added a new IP-MS dataset from plants subjected to cold (new SuppFig2).** The results validated many of the interactions observed in warm. This is described in the results section in lines 123-124. The VRN1 interactions in the cold reproducibly show proteins involved in histone deacetylation, chromatin remodelling, Polycomb silencing, and nuclear envelope associated with VRN1. All proteins shown in the new SuppFig2 are enriched in the VRN1-FLAG IP relative to IP in the WT without FLAG. We also noticed proteins that were not enriched in VRN1 IP in the warm, but the function of those is the same. HD2C is an example of a protein strongly detected after VRN1 IP in the warm but is much less enriched in the cold. That is consistent with the observation that HD2C is specifically degraded in a cold-dependent manner (Park *et al.*, 2018 PNAS 115: E5400-09).

2. MNase-seq plots were shown in Figure 3C and 3D. However, these figures did not show any visible changes between different samples. A statistical analysis was performed in Figure 3E. However, it is weird that there are three values for the first three samples but four values for the last sample (vrn1 FRI 1W cold). The variant of the last sample is clearly higher than those of the three other ones. All the four values should be included in the statistical analysis.

The Reviewer had concerns about the differences observed in the previous MNase-seq plots. This was shared with Reviewer 1 and has been addressed in the new version of the manuscript. As discussed in detail under Reviewer 1 point 1, we now provide new analyses and visually clearer plots. The **new Fig3D** shows a more straightforward representation of nucleosome position as a function of sequence. It is now clearer that the +1 nucleosome in the WT in the warm has one predominant position (one major peak), while in the cold, a stronger signal is detected closer to the TSS, representing the cold-induced +1 repositioning. It is also visually clearer that this cold-induced repositioning is affected in the *vrn1* mutant. As this is only a visual representation of the data, it has been quantified in the **new Fig3E**. This shows the ratio of TSS-proximal +1 (repositioned) to TSS-distal +1 (not repositioned) in the cold normalised to the warm. We have also tested for significant differences. While the cold-induced repositioning is statistically significant in the WT, in the *vrn1* mutant, it is not. An alternative analysis quantifies the mean nucleosome centre position (+/- standard deviation) without normalisation, which has been added in the new plots in **SuppFig5A, B**. The results again show that the +1 nucleosome's centre position is significantly closer to the TSS after cold in the WT but not in the mutant. They also show significant differences between WT and *vrn1* in the cold. We believe these new plots have strengthened our conclusions.

Although there are significant differences between warm and cold, and WT and mutant, in respect to the +1 nucleosome position, we do not observe large general changes in nucleosome distribution along the entire *FLC* locus, which is in line with part of the Reviewer's comment and was a message we wanted to convey with Fig3C. As we describe in the text (lines 185-194), there are no obvious changes in the overall nucleosome distribution or occupancy despite changes in gene expression. We also draw the reader's attention to the diverse signal pattern across the locus in terms of positioning and DNA length, hence the complex plot in Fig3C. We believe this reflects biophysical differences between the different nucleosomes regarding their positions relative to each other, breathing dynamics, composition, or other aspects that contribute to their sensitivity to MNase digestion.

The Reviewer also shows concern regarding the number of replicates. As mentioned in the methods section (lines 627-628), we have performed the targeted MNase-seq using three replicates for samples in the warm (WT and mutant) and four replicates for samples in the cold. That is because more plant material was obtained after one extra week of cold exposure. No replicates have been omitted; the values for all independent biological replicates are shown in the quantification plots as individual points (**Fig3E**). In **SuppFig5**, all replicates contributed to the mean +/- standard deviation. The visual distribution in **Fig3D** shows a cumulative number from all biological replicates. The individual distribution of nucleosomal DNA position and length at the 5' end of *FLC* for all biological replicates is provided in the **SuppFig6**.

3. FRAP was used for testing the impact of *vrn1* and cold on chromatin dynamics, indicating that both H2A.Z-containing chromatin and H3.3-containing chromatin are more dynamic in *vrn1* than in WT. Because the *vrn1* mutant specifically showed flowering time alteration but had no other effect. It is difficult to understand why the *vrn1* mutation has significant effect on chromatin dynamics but does not affect genome-wide gene expression or development. Moreover, an explanation should be provided to show why H2A.Z and H3.3 but not canonical H2A or H3 were used in the analysis. At least, the canonical histones should be used as a negative control.

The biological function of VRN1 is something we did not fully address in the previous version of the manuscript. This has been changed in the revised version. As we now comment on the text (lines 141-146), the global impact of VRN1 on histone dynamics (as revealed by the FRAP experiments) is not surprising given that VRN1 does not only affect flowering time via *FLC*. Previous work showed that VRN1 overexpression causes accelerated flowering independent of *FLC* and vernalization, in part due to ectopic expression of *FT* and *AGL20* genes (ref 37). Additionally, VRN1 protein was shown to broadly associate with all mitotic chromosomes during cell division (ref 38). Further work has proposed that VRN1 regulates essential developmental processes, although that is masked by redundancy. Using a dominant repressor tag to overcome genetic redundancy, VRN1 was found to be essential for Arabidopsis development (ref 41).

H2A.Z and H3.3 are known to affect chromatin dynamics and Polycomb silencing, presumably by affecting properties such as nucleosome stability and turnover. Therefore, they could potentially mediate the impact of VRN1 on the chromatin. The interaction of VRN1 with many chromatin regulators, including chromatin remodelling complexes, and its broad association with chromatin during mitosis could explain the global impact on H2A.Z and H3.3 dynamics. Performing additional FRAP experiments for other histones would require obtaining the transgenic tagged-histones in a background with the FRI introgression and *vrn1* mutation. Obtaining such lines in these late-flowering mutants would extend much beyond the timeframe of revision of this manuscript. Instead, to address whether the impact of the *vrn1* mutation is general for both histone variants or context-specific, we have performed additional ChIP-qPCR experiments. New data has been added to the **SuppFig3B**, showing H3.3 levels across the *FLC* locus in the warm or after six weeks of cold in the WT and mutant. The results show that, at *FLC*, even though *vrn1* changes H2A.Z both in the warm and cold (predominantly at the 5' end), the impact on H3.3 is only observed in the cold (largest at the 3' end). This suggests that despite the general increase in histone dynamics, the impact of VRN1 on each variant depends on the target region.

4. The nucleosome position was thought to be affected by cold and by *vrn1* mutation. However, the results shown in Figure 3D and 3E seem not convincing.

The Reviewer had concerns regarding the differences observed in the previous MNase-seq plots, which have been pointed out in point 2 and shared with Reviewer 1 point 1. We have provided new analyses and plots, and a full response that can be found under Reviewer 1 point 2 and in response to Reviewer 1 point 1.

5. In Figure 3A, five loci were selected for the detection of solubilised chromatin. Although the effect of cold treatment and *vrn1* mutation on H3K27me3 and H2Aub were detected specifically in the regions nearby the transcription start site and intragenic regions (Figure 1), the effect of cold treatment and *vrn1* mutation on the solubilised chromatin did not show any locus specificities. Especially, the cold treatment significantly reduced the solubilised chromatin level at all the five tested loci (Figure 3A). I wonder whether the experiment can really detect the chromatin state *in vivo*.

We apologise if this was not clear in the previous version of the manuscript. We did assess the effects of cold and *vrn1* on chromatin solubility over the 5' end of *FLC* where we observed changes in H3K27me3, H2Aub, and H2A.Z. This was done using five different qPCR amplicons, all mapping in the 5' region of *FLC* and not to five different loci. We made this clearer in the new version of the manuscript (see lines 152-153 of the text with track changes).

6. Figure 4 shows a model for the proposed effect of nucleosome dynamics on chromatin structure change at *FLC* locus, while Figure 5 illustrates the simulation of *FLC* chromatin structure based on the DNA sequence and nucleosome position. The model and the simulations are totally based on the results shown in Figure 3. As indicated above, the results shown in Figure 3 need to be further confirmed using additional data.

We have corroborated the results in Fig3 with new analyses and plots as discussed under point 2 and in response to Reviewer 1 point 1. We would also like to point out that the model in **Fig4** is based on data from all the figures and not just Fig3. The model represents the impact of VRN1 on the local nucleosome dynamics underlying the *FLC* transcriptional shutdown and stable silencing. The model shows the constitutive activity of VRN1 in limiting H2A.Z both in the warm and in the cold, as revealed by ChIP-qPCR for VRN1 (**Fig2A**), ChIP-qPCR for H2A.Z (**Fig1D**), and other links to H2A.Z, such as the epistatic interaction with *arp6* (**SuppFig1A-C**), the association with INO80 *in vivo* (**Fig2D** and **SuppFig2**), and the impact on H2A.Z histone dynamics (**Fig2E**). This is coupled with the cold-induced repositioning of the +1 nucleosome (**Fig3**). The model also shows a scenario where VRN1 function has been impaired, which leads to hyper-accumulation of H2A.Z, compromised nucleosome positioning, high nucleosome dynamics, and defects in the *FLC* transcriptional shutdown (**Fig1A**), and Polycomb silencing (**Fig1B,C**). The model also attempts to represent a potential impact from the local nucleosome dynamics on the 3D chromatin structure, which was the premise for the computational simulations.

The simulations are also not entirely based on the results in Fig3. The simulations are based on the mechanical properties of the DNA, which are inherent to the *FLC* sequence, and electrostatic and hydrophobic interactions, which are the basis for nucleosome-nucleosome and nucleosome-DNA contacts. The only input of data from Fig3 to the simulations was the most frequent positions observed

in the WT in the warm and in the cold. That was used as the nucleosome positions at the start of the simulations. The simulations also integrate H2A.Z incorporation as informed by multiple data shown in various figures. This attempted to characterise the chromatin structure, properties and behaviour at *FLC*, in a state representing loss of *VRN1* function. The simulations confirm the importance of the local nucleosome dynamics to the chromatin structure at *FLC*, as proposed in the model in **Fig4**. The simulations also provided further insights into potentially important structural aspects underlying the switch from an active transcription state to one that can be Polycomb-silenced.

7. In line 235, "VRN1 genetically depends on ARP6 to silence FLC, implicating H2AZ eviction as a functional requirement for FLC silencing in the cold". This statement is not correct. ARP6, as a SWR1 complex, promotes the deposition of H2A.Z on chromatin. The ARP6-dependent deposition of H2A.Z on the FLC locus enhances FLC expression but not for silence FLC. In factor, VRN1 and ARP6 have opposing effects on FLC expression.

Thanks for pointing that out. We have rephrased that in the Discussion section (lines 335-348). The genetic analyses indicate that VRN1 functionality is no longer important for *FLC* silencing in an *arp6* mutant, with defective H2A.Z incorporation. A comment on this was also made in the Results section (lines 98-103).